# Fear memory recall involves hippocampal somatostatin interneurons

**Krisztián Zichó**[1,2], **Katalin E. Sos**[1], **Péter Papp**[1], **Albert M. Barth**[1], **Erik Misák**[1], **Áron Orosz**[1], **Márton I. Mayer**[1,2], **Réka Z. Sebestény**[1], **Gábor Nyiri**[1]*

1 Laboratory of Cerebral Cortex Research, Institute of Experimental Medicine, Budapest, Hungary, 2 János Szentágothai Doctoral School of Neurosciences, Semmelweis University, Budapest, Hungary

* nyiri@koki.hu

## Abstract

Fear-related memory traces are encoded by sparse populations of hippocampal principal neurons that are recruited based on their inhibitory–excitatory balance during memory formation. Later, the reactivation of the same principal neurons can recall the memory. The details of this mechanism are still unclear. Here, we investigated whether disinhibition could play a major role in this process. Using optogenetic behavioral experiments, we found that when fear was associated with the inhibition of mouse hippocampal somatostatin positive interneurons, the re-inhibition of the same interneurons could recall fear memory. Pontine nucleus incertus neurons selectively inhibit hippocampal somatostatin cells. We also found that when fear was associated with the activity of these incertus neurons or fibers, the reactivation of the same incertus neurons or fibers could also recall fear memory. These incertus neurons showed correlated activity with hippocampal principal neurons during memory recall and were strongly innervated by memory-related neocortical centers, from which the inputs could also control hippocampal disinhibition in vivo. Nonselective inhibition of these mouse hippocampal somatostatin or incertus neurons impaired memory recall. Our data suggest a novel disinhibition-based memory mechanism in the hippocampus that is supported by local somatostatin interneurons and their pontine brainstem inputs.

## Introduction

Understanding how fear memories are acquired and recalled in the brain is a fundamental goal in neuroscience. Recent studies have revealed that different memory traces are encoded by different subpopulations of principal neurons in hippocampal formation [1–3]. These subpopulations of neurons are called engram cells [4–6]. The hippocampal dentate gyrus (DG) is especially important for encoding the contextual components of memories [7,8]. Experimental inhibition of DG engram cells formed during memory acquisition impaired contextual fear memory recall [9–11], whereas experimental reactivation of fear memory engram cells can recall fear memories even in a novel, neutral environment [12–14]. However, the mechanism of allocating DG granule cells (GCs) to an engram cell population remains unclear. Current models suggest a near-random process orchestrated by an excitatory drive from the entorhinal

**Funding:** This work was supported by the Hungarian National Research, Development and

Innovation Office (OTKA K119521); the Frontline Research Excellence Program, by the Hungarian National Research, Development and Innovation Office (NRDI Fund 133837), the Hungarian Brain Research Program [NAP2.0 (2017-1.2.1-NKP-2017-00002) and NAP3.0 (NAP2022-I-1/2022)], European Union project RRF-2.3.1-21-2022-00004 within the framework of the Artificial Intelligence National Laboratory; European Union project RRF-2.3.1-21-2022-00011 within the framework of the Translational Neuroscience National Laboratory to G.N. The National Academy of Scientist Education Program of the National Biomedical Foundation under the sponsorship of the Hungarian Ministry of Culture and Innovation (FEIF/646-4/2021-ITM_SZERZ) to G.N and R.Z.S. The New National Excellence Program of the Ministry of Innovation, Hungary, UNKP-20-3-SE-31 and UNKP-21-3-SE-9 and the Semmelweis 250+ Excellence PhD Fellowship, EFOP-3.6.3-VEKOP-16-2017-00009 to K.Z. The National Research, Development and Innovation Office, Hungary, FK129019 to A.M.B. The funders had no role in study design, data collection and analysis, decision to publish, or preparation of the manuscript.

**Competing interests:** The authors have declared that no competing interests exist.

**Abbreviations:** AAV, adeno-associated virus; ACC, anterior cingulate cortex; ArchT, archaerhodopsin T-3; ChR2, channelrhodopsin 2; CNO, clozapine-N-oxide; DG, dentate gyrus; DREADD, designer receptors exclusively activated by designer drugs; eOPN3, encephalopsin 3; FG, FluoroGold; GABA, γ-aminobutyric acid; GC, granule cell; HSA, human serum albumin; hM4Di, modified human M4 muscarinic receptor; mPFC, medial prefrontal cortex; NI, nucleus incertus; PV, parvalbumin; RSC, retrosplenial cortex; SOM, somatostatin; vGAT, vesicular GABA transporter; vGluT1, vesicular glutamate transporter 1.

cortex (that carries context-related information), modulations from local interneurons, and external inputs [8,15–17]. However, memory recall requires the precise reactivation of prese-lected engram cells [9–11,18], for which the rather unspecific entorhinal input does not seem particularly suitable [15]. Principal cells that are more excitable during a given salient event seem to be allocated to the memory engram of that event more effectively [10,19–21], whereas experimental inhibition of the excitability of naturally formed engram cells inhibits contextual memory consolidation and recall [10,11,20,22,23].

The excitability of DG GCs can be increased by inhibiting their dendritic inhibitory inter-neurons [16,24,25]. This type of disinhibition is an important modulatory process in several brain areas [26–30]. Dendrite-targeting somatostatin positive inhibitory interneurons (SOM cells) [unlike, e.g., parvalbumin (PV) positive interneurons] are known to be especially suitable for modulating the excitability of principal cells both in the DG and in the CA1 regions of the hippocampus [16,31,32]. SOM cells have also been suggested to modulate memory formation [16,25,32–34], reactivation of the principal cells during consolidation [22,23], or the number of engram cells [16,34]. Indeed, axonal arborizations of these SOM cells are uniquely associ-ated with the hippocampal arborizations of the excitatory axons from the entorhinal cortex that carries sensory-related information onto hippocampal principal cell dendrites [8,35–38]. This specific axonal association makes them ideally suitable for modulating hippocampal memories. In addition, SOM cells show unique molecular changes during memory consolida-tion, similar to those found only in principal neurons [39]. We and others have previously demonstrated that hippocampal SOM cells are selectively targeted by the brainstem nucleus incertus (NI) [34,40]. NI can influence memory acquisition [34], whereas its inhibition can impair memory recall [40–42].

Here, we discovered that SOM cells could form a key relay network of a previously unrecog-nized, disinhibition-based hippocampal memory formation and recall mechanism. We found that fear memory can be encoded into a subset of GCs by releasing them from the inhibition of a subset of SOM cells (but not PV cells) during a fearful event. This up-regulation of the excitability of GCs was so efficient that we could effectively recall those fear memories even in a novel environment, only by re-inhibiting the same subset of SOM cells. In addition, we found that hippocampal CA1 SOM cells can similarly support the acquisition and recall of fear memories. We also discovered that the precisely timed inhibition of SOM cells could originate from the inhibitory γ-aminobutyric acid (GABA)-releasing cells of the pontine brainstem NI. By activating either these NI GABAergic cell bodies or their axonal fibers in the hippocampus, we stored fear memories during a fearful event, and we could effectively recall those fear mem-ories even in a novel environment, only by re-activating the same subset of NI GABAergic cells or their hippocampal fibers. Furthermore, we found that this pathway is vital for contex-tual fear memory recall because inhibiting a subpopulation of hippocampal SOM or NI GABAergic cells during the memory recall period significantly impaired fear memory recall. We have also discovered that this disinhibition-based memory mechanism could be supported by the massive and selective innervation of NI from key memory-related neocortical centers, such as the anterior cingulate cortex (ACC), whose terminals in the NI can also control hippo-campal inhibition in vivo.

## Results

### Specific inhibition of DG SOM cells can recall fear memory

During memory acquisition, more excitable DG GCs are more likely to form an engram [10,16]. This needs to be controlled tightly because the recall may later require increasing the excitability of the same GCs. Dendrite-targeting interneurons seems ideal for this task.

Inhibition of a subset of DG SOM cells during memory acquisition could make their target GCs more excitable [16], so we hypothesized that re-inhibiting the same SOM cells later could recall the memory. To test this, we used a fear conditioning paradigm, where memory recall can be tested by analyzing freezing behavior that indicates fear memory recall. First, we used encephalopsin 3 (eOPN3), which effectively inhibits neurons in response to light activation [43]. We employed Cre-dependent adeno-associated virus (AAV) constructs to express these opsins in target cells. The in vivo expression of all viruses in this study was specific to Cre-expressing cells, including that of the AAV1-eOPN3-mScarlet construct (S1F and S1G Fig). First, we injected Cre-dependent eOPN3-containing AAVs ("eOPN3-mice") or Cre-dependent control AAVs ("CTRL-mice") into the DG of SOM-Cre mice bilaterally, and then, we implanted optic fibers above the injection sites (Fig 1A, Materials and methods). After handling, the mice were placed into environment "A" to record their baseline behavior for 3 min (light OFF, Fig 1A). Then, the light was switched ON for 3 min (light ON, Fig 1A). Neither CTRL nor eOPN3-mice showed behavioral changes during these periods (S1C Fig), indicating that inhibition of DG SOM cells in itself cannot induce freezing behavior. The next day, all mice were placed into environment "B," where they received foot shocks. eOPN3-mice were separated into 2 groups. CTRL-mice and some eOPN3-mice (called "Associated eOPN3-mice") received light illumination aligned to the foot shocks (Fig 1A). The other eOPN3-mice (called "Not associated eOPN3-mice") did not receive light illuminations (Fig 1A). Next, we investigated whether fear memories could be recalled by re-inhibiting DG SOM cells, which in return could reactivate the memory encoding cell assembly. This could not be tested in the conditioned environment "B" because that would have recalled the fear memories naturally, therefore, on the next day, the mice were placed in a novel environment "C" to investigate whether light illuminations themselves recall memory. First, we recorded behavior without light illumination (2 min light OFF period). Initially, as expected, mice showed some fear behavior because it was not possible to build a novel environment that is completely different from environment "B" (see Materials and methods). However, after the mice realized that they are in a novel environment, this fear quickly decreased during this initial period (S1A Fig) and it did not differ among the groups (Figs 1B and S1C). Then, after 2 min, we illuminated the DGs for 3 min (light ON period). By that time, the mice had already habituated to this context, and the fear behavior of CTRL and "Not associated eOPN3-mice" diminished during the light ON period (Figs 1B–1D, S1A and S1C). However, "Associated eOPN3-mice" significantly increased their freezing behavior (Figs 1B–1D, S1A and S1C). In some "Associated eOPN3-mice," the freezing behavior was observed immediately, whereas for the whole group, the increase was already significant within 30 s (S1B Fig). This shows that these mice recalled their fear memories encoded in the DG during contextual conditioning. Finally, after switching off the light for 2 min (light OFF period), fear levels of "Associated eOPN3-mice" but not CTRL or "Not associated eOPN3-mice" decreased significantly, and fear responses among the 3 groups were again not different (Figs 1B, S1A and S1C). Next, we tested whether precise temporal association between the inhibition of DG SOM cells and the mild foot shocks plays a role in the efficiency of the recall of fear memories later. We found that the associations were significantly more effective when optogenetic inhibition was precisely aligned to the aversive stimulus, rather than presented after a 60 s temporal shift (S2A–S2C Fig). These data show that temporally and spatially specific inhibition of a subset of SOM cells can induce fear memory recall. However, it also shows that a nonspecific inhibition of SOM cells, even after a powerful, recent fearful event, cannot recall memory unless the same subset of SOM cells primes their target GCs during memory acquisition. Such an effective memory recall could be achieved by a small subset of SOM cells because: (i) the opsins could be activated only within the intersection of the region of the virus injection and the light-cone from the optic fibers; (ii) these areas

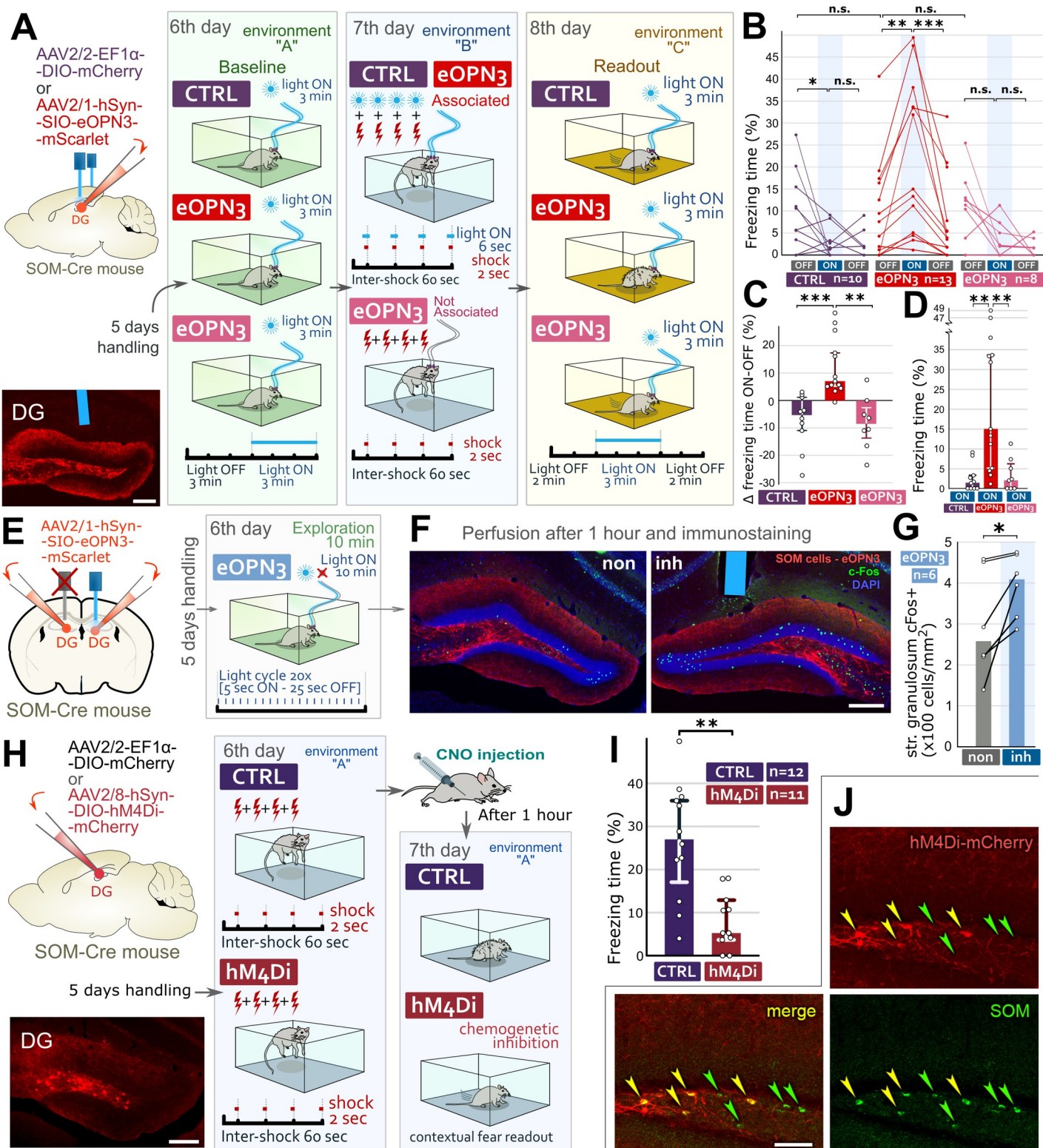

**Fig 1. Re-inhibition of DG SOM cells inhibited during memory acquisition recalls fear memory.** (A) After infecting DG SOM cells with inhibitory opsin-containing ("eOPN3 mice") or control AAVs ("CTRL mice") bilaterally, we implanted optic fibers over DGs. Representative image: injection site and optic fiber position (blue). Scale bar: 200 μm. Day 6: baseline freezing behavior of mice without and with light illumination. Day 7: CTRL and "Associated eOPN3-mice" received foot shocks aligned with light illumination, "Not associated eOPN3-mice" received foot shocks but no light. Day 8, in environment C: 2 min OFF 3min ON 2 min OFF light cycle. Only "Associated

eOPN3 mice" (red) could recall fear memory. (B) Individual percentages of time spent with freezing behavior during the light OFF-ON-OFF cycles on day 8. (C) Changes in freezing behavior between the first light OFF and ON periods (medians and interquartile ranges). (D) Freezing time differences during the light ON periods on day 8 (medians and interquartile ranges). (E) Bilateral infection of DG SOM cells with inhibitory opsin and bilateral optic fiber implantation. After handling, the mice spent 10 min in a novel environment, where only 1 side of the DG was light illuminated. (F) Representative images: c-Fos (green) labeling of non-illuminated (non) and illuminated side (inh, where SOM cells are inhibited) of the DG, 1 h after exploration (optic fiber is blue). Scale bar: 200 μm. (G) Differences in the density of c-Fos positive cells (cells/mm2) in the DG GC layer in non-illuminated (non) and light disinhibited (inh.) side of the DG in 6 eOPN3 mice (medians and interquartile ranges). (H) We infected DG SOM cells with chemogenetic inhibitory ("hM4Di mice") or control AAVs ("CTRL mice") bilaterally. Image shows a representative injection site. Scale bar: 200 μm. After handling, the mice received 4 foot shocks on day 6. Day 7: all mice received CNO 1 h before contextual memory readout. (I) Contextual fear of hM4Di mice almost completely diminished compared to CTRL mice on day 7 (medians and interquartile ranges). (J) hM4Di-staining (red) is present only in a subpopulation (yellow) of DG SOM cells (green). Scale bar: 50 μm (for details, see S1 Extended Data). The data underlying this figure can be found in S1 Data. AAV, adeno-associated virus; DG, dentate gyrus; eOPN3, encephalopsin 3; GC, granule cell; hM4Di, modified human M4 muscarinic receptor; SOM, somatostatin.

are relatively small compared to the whole hippocampus; and (iii) we found that only about 55% of SOM cells were infected by the AAVs below the tip of the optic fibers (S1G Fig). On the last day, we confirmed that both "Associated" and "Not associated" eOPN3-mice could recall their natural contextual fear memories equally well in environment "B" (S1D Fig).

## Inhibition of DG SOM cells is sufficient to activate DG GCs

We also confirmed that the inhibition of a subset of DG SOM cells is sufficient to increase the number of activated GCs in DG significantly. DG SOM cells were infected bilaterally with eOPN3-containing inhibitory AAVs, and we implanted 2 optic fibers over the injection sites (Fig 1E). Later, mice were placed into a novel environment for 10 min, where we inhibited SOM cells only on 1 side of the DG ("inh. side") by light. The other side was not illuminated ("non side," Fig 1E). An hour later, immediate early gene c-Fos expression showed that significantly more GCs were activated on the illuminated sides (Fig 1F and 1G), indicating that inhibiting SOM cells via eOPN3 is sufficient to activate a subpopulation of GCs.

## DG SOM interneurons are necessary for natural contextual fear memory recall

We also demonstrated that DG SOM cells are required for natural contextual fear memory recall using a DREADD (designer receptors exclusively activated by designer drugs, hM4Di)-containing AAV-based chemogenetic vector that expresses mutant receptors that inhibit cells upon binding to clozapine-N-oxide (CNO). We infected DG SOM cells in SOM-Cre mice with chemogenetic inhibitory ("hM4Di mice") or control AAVs ("CTRL mice") bilaterally (Fig 1H). On day 6, mice received foot shocks in environment "A" (Fig 1H, see Materials and methods for details). The next day, both hM4Di and CTRL-mice received injections of CNO, and 1 h later, they were placed back into environment "A." hM4Di-mice showed significantly lower fear behavior (almost none) compared to CTRL-mice (Fig 1H and 1I) and even that behavior developed slower (S1E Fig). Then, using immunohistochemistry, we found that this effect was achieved by inhibiting only about 60% of the cells in solely the dorsal DG in hM4Di mice (Fig 1J). This demonstrated that hippocampal SOM cells are required for natural contextual fear memory recall because the nonspecific inhibition of even a small subpopulation of these cells impairs memory recall in the conditioned environment.

## Even the specific inhibition of DG PV cells cannot recall fear memory

Perisomatic PV interneurons, one of the most numerous interneuron populations in the hippocampus, effectively regulate the firing rate rather than the excitability of GCs [16,44]. Here, we demonstrated that, unlike DG SOM cells, the re-inhibition of DG PV cells inhibited during memory acquisition could not recall fear memory. We injected Cre-dependent

eOPN3-containing ("eOPN3-mice") or control AAVs ("CTRL-mice") into the DG of PV-Cre mice bilaterally and implanted 2 optic fibers over the injection sites (Fig 2A). After handling, we performed an optogenetic inhibition-based memory recall paradigm (Fig 2A) similar to that used with SOM-Cre mice. Briefly, on day 6, baseline recordings (without and with light) demonstrated no changes in fear behavior, indicating that inhibition of DG PV cells alone cannot induce fear (S2D Fig). The next day, the mice received foot shocks together with light illumination of the DG (Fig 2A). The next day, mice were tested using the 2-3-2 min light OFF-ON-OFF paradigm in a novel environment. No mice showed increased freezing behavior, and eOPN3-mice did not differ from the controls (Figs 2B–2D and S2D). Therefore, unlike DG SOM cells, the specific re-inhibition of DG PV cells could not recall fear memory.

## Specific inhibition of CA1 SOM cells can also recall fear memory

Previously, the inhibition of CA1 SOM cells during foot shocks prevented contextual fear memory recall [32,34]. Here, we investigated whether that memory trace was lost or it had merely remained silent [5] and could have been recalled by the re-inhibition of the same subset of CA1 SOM cells. We infected CA1 SOM cells with inhibitory archaerhodopsin T-3 [ArchT, [45]]-containing ("ArchT-mice") or control AAVs ("CTRL-mice") bilaterally and implanted 2 optic fibers over the injection sites (Fig 2E). After handling, we performed the same inhibition-based memory recall paradigm (Fig 2E) similar to that used with PV-Cre mice. On day 6, the baseline recordings (without and with light) demonstrated no changes in fear behavior, indicating that the inhibition of CA1 SOM cells cannot induce fear (S2E Fig). The next day, the mice received foot shocks associated with light illumination of the CA1 area (Fig 2E). The next day, mice were tested using the 2-3-2 min light OFF-ON-OFF paradigm in a novel environment. No difference was found during the first light OFF period (Figs 2F and S2E). Then, illuminating the CA1 areas for 3 min (light ON period) induced significantly higher fear behavior of ArchT-mice compared to CTRL-mice (Figs 2F–2H, S2E and S2F). After switching off light illumination (second light OFF period), the fear responses of CTRL-mice did not change, but those of ArchT-mice decreased significantly and became like those of the CTRL-mice (Figs 2F, S2E and S2F). These data demonstrated that (like DG SOM cells) re-inhibition of CA1 SOM cells inhibited during memory acquisition could also recall fear memory.

## NI GABAergic fibers can activate GCs by disinhibition via SOM cells

Brainstem NI GABAergic fibers are known to target DG SOM cells [34]. Here, we tested whether they could indirectly activate DG GCs via DG SOM interneurons. First, using a combination of flippase- and Cre-dependent AAVs in double transgenic mice (Fig 3A), we found that NI GABAergic fibers not only selectively target hippocampal SOM cells [34], but also they directly and strongly innervate DG SOM cells perisomatically in the hilus of DG [Figs 3B and S4A, at least 87% (509/583) of virally labeled NI axonal terminals formed synapse-specific gephyrin-labeled synaptic contacts with DG SOM cells ($n$ = 2 mice)]. Using monosynaptic retrograde rabies tracing, we found no other extra-hippocampal inputs that would specifically target only DG SOM cells (S3A and S3B Fig). Although rabies tracing is less efficient between distant brain regions, we could demonstrate that the well-known nonspecific cholinergic and GABAergic inputs from the basal forebrain and some nonspecific glutamatergic inputs form the median raphe can target DG SOM cells as well. In addition, DG SOM cells also received local hippocampal inputs (S3C–S3G Fig).

To demonstrate the disinhibition of GCs, we infected NI GABAergic cells with light-sensitive excitatory opsin-containing ("ChR2-mice") or control AAVs ("CTRL-mice") in vesicular GABA transporter (vGAT)-Cre mice (Fig 3C) and implanted optic fibers over the DG

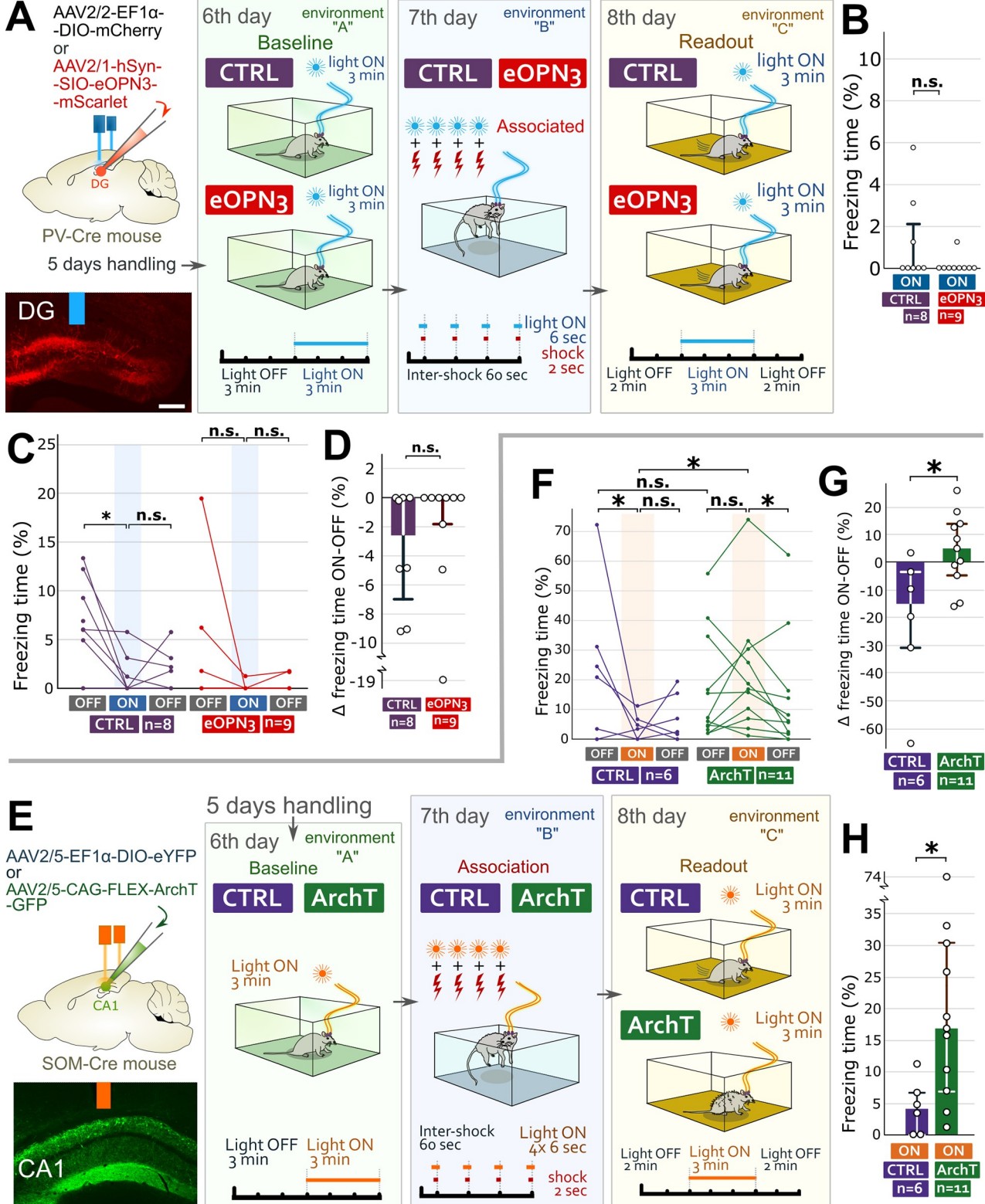

**Fig 2. Unlike DG PV cells, the re-inhibition of CA1 SOM cells inhibited during memory acquisition recalls fear memory.** (A) After infecting DG PV cells with inhibitory opsin-containing ("eOPN3 mice") or control AAVs ("CTRL mice") bilaterally, we implanted optic fibers over DGs. A representative image: injection site and optic fiber position (blue). Scale bar: 200 μm. Day 6: baseline freezing behavior of mice without and with light illumination. Day 7: All mice received foot shocks aligned with light illumination. Day 8 in environment C: 2 min OFF 3 min ON 2 min OFF light

cycle demonstrated that (unlike DG SOM cells) optical re-inhibition of DG PV cells could not recall fear memory. (B) No differences in freezing between PV-Cre mouse groups during the light ON period on day 8 (medians and interquartile ranges). (C) Individual percentages of time spent with freezing behavior during the light OFF-ON-OFF cycles on day 8. (D) Changes in freezing behavior between the first light OFF and ON periods (medians and interquartile ranges). (E) After infecting CA1 SOM cells with inhibitory opsin-containing ("ArchT mice") or control AAVs ("CTRL mice") bilaterally, we implanted optic fibers over the CA1 areas. Representative image: injection site and optic fiber position (orange). Scale bar: 200 μm. Day 6: baseline freezing behavior of mice without and with light illumination. Day 7: All mice received foot shocks precisely aligned with light illumination. Day 8 in environment C: 2 min OFF 3 min ON 2 min OFF light cycle demonstrated that optical re-inhibition of CA1 SOM cells in itself could recall fear memory. (F) Individual percentages of time spent with freezing behavior during the light OFF-ON-OFF cycles for each SOM-Cre mouse on day 8. (G) Significant difference in the changes in freezing behavior between the first light OFF and ON periods (medians and interquartile ranges). (H) Freezing time difference between SOM-Cre mouse groups during the light ON period on day 8 (medians and interquartile ranges; for details, see S2 Extended Data). The data underlying this figure can be found in S1 Data. AAV, adeno-associated virus; ArchT, archaerhodopsin T-3; DG, dentate gyrus; eOPN3, encephalopsin 3; PV, parvalbumin; SOM, somatostatin.

bilaterally (Fig 3C). Later, mice were placed into a novel environment for 10 min (Fig 3C), where we light-stimulated NI GABAergic fibers only on 1 side of the DG. The other side was not illuminated (Fig 3C). Forty minutes later, immediate early gene c-Fos expression showed that significantly more GCs were activated on the illuminated sides of ChR2-mice (but not CTRL-mice, Fig 3D and 3E), indicating that a subpopulation of NI fibers can indirectly activate a subpopulation of GCs.

To explore this functional connectivity in vivo between NI GABAergic neurons and the DG, we performed in vivo electrophysiological recordings from vGAT-Cre mice injected with Cre-dependent ChR2-expressing viruses into the NI. DG unit activity was recorded by using two 128 channel silicone probes parallel with optogenetic activation of NI (Fig 3F). The laser stimulation ON-OFF periods were alternated during the recordings. Although several putative excitatory cells have changed their activity during the light ON periods at least 3 out of 54 (5.6%) increased their activity consistently and significantly during laser stimulations (Fig 3G and 3H). Although optical stimulation cannot mimic the original network activity, this ratio is close to the low ratio of the cells that participate in any given event-related cell assembly. In case of the inhibitory cells, from the 170 putative inhibitory units, at least 5 (2.9%) decreased their activity consistently and significantly upon laser stimulation (Fig 3I). These data shows that NI can influence network activity of the DG by inhibiting interneurons and disinhibiting GCs.

## Specific activation of NI GABAergic fibers can recall fear memory

Previously, we demonstrated that GABAergic fibers of the NI are activated by salient events in the hippocampus during memory acquisition [34]. Because NI GABAergic cells strongly innervate DG SOM cells, here we investigated whether reactivation of the same subset of NI GABAergic fibers can also recall fear memories. Therefore, we performed a fear conditioning paradigm (Fig 4A), similar to that we performed above. We infected NI GABAergic cells with light-sensitive excitatory opsin-containing ("ChR2-mice") or control AAVs ("CTRL-mice") in vGAT-Cre mice and implanted optic fibers over the DG bilaterally (Fig 4A). On day 6, the baseline recordings (without and with light) showed no changes in fear behavior, indicating that activation of NI GABAergic fibers cannot induce fear (S5A Fig). The next day, mice received foot shocks aligned with the light illumination of the DGs (Fig 4A). The next day, mice were tested using the 2-3-2 min light OFF-ON-OFF paradigm in a novel environment. No difference was found during the first light OFF period (Figs 4B and S5A). Then, illumination of the DGs for 3 min (light ON period) caused a significantly increased fear behavior of ChR2-mice (Figs 4B–4D, S5A and S5B). In some ChR2-mice, freezing behavior was observed immediately, whereas, for the whole group, the increase became significant within 1 min (S5B). After switching off the light illumination (second light OFF period), the fear responses

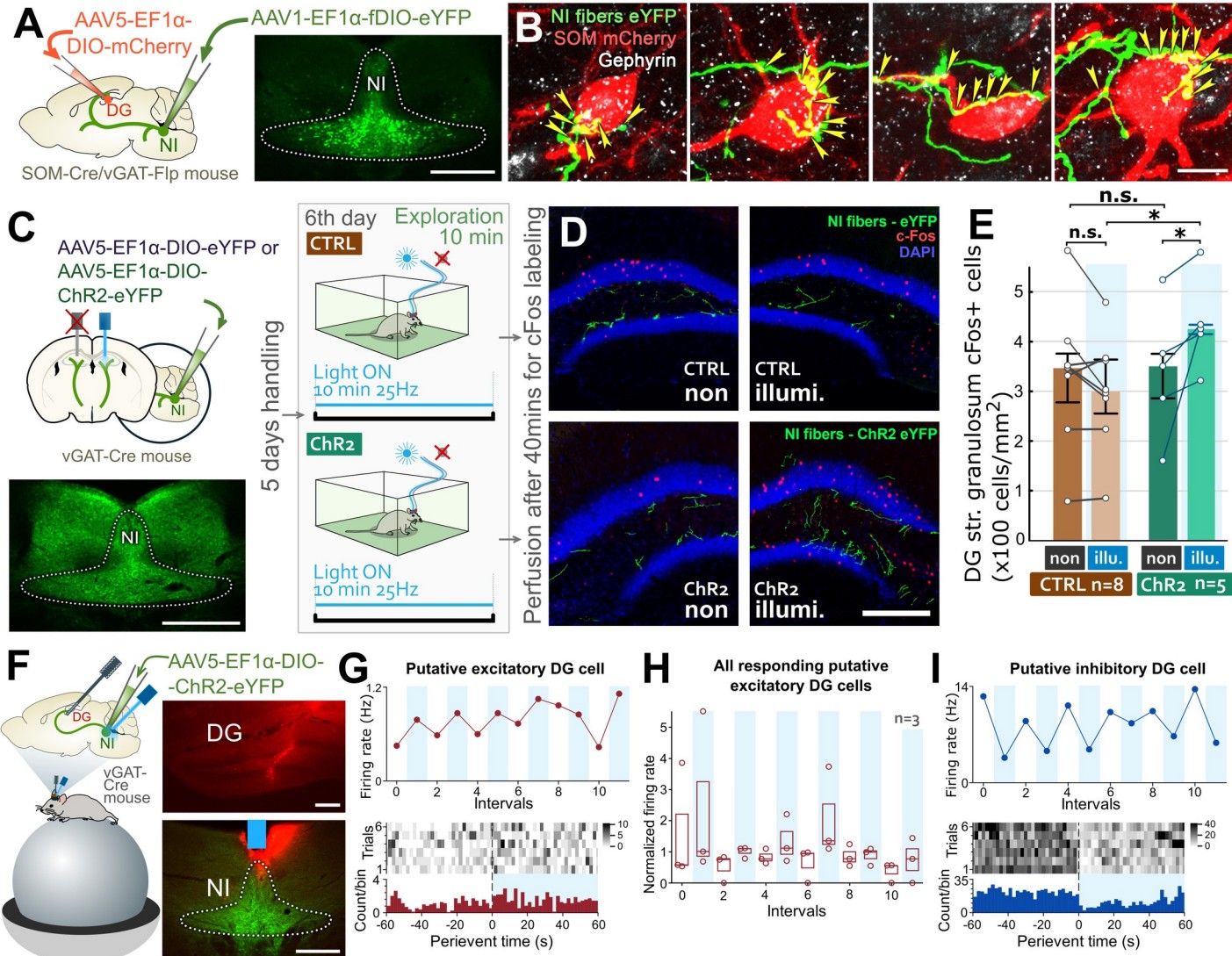

**Fig 3. NI GABAergic cells can disinhibit DG GCs via inhibiting DG SOM interneurons.** (A) We infected DG SOM cells with Cre-dependent tracer AAV and NI GABAergic cells with flippase-dependent tracer AAV in double transgenic SOM-Cre/vGAT-Flp mice (*n* = 4 mice). The representative image shows the injection site in NI. Scale bar: 200 μm. (B) SOM cells (red) receive multiple gephyrin-labeled (white) perisomatic synapses (yellow arrowheads) from NI GABAergic fibers (green). Scale bar: 10 μm. (C) Infection of NI GABAergic cells with either control ("CTRL mice") or excitatory opsin-containing AAVs ("ChR2 mice") and bilateral optic fiber implantation over the DG. The representative image shows the injection site in NI. Scale bar: 200 μm. After handling, the mice spent 10 min in a novel environment, where only 1 side of the DG was light illuminated. (D) Representative images show c-Fos labeling (red) of non-illuminated (non) and illuminated side (illumi.) of the DG, 40 min after exploration. Images show both sides for both the CTRL- and ChR2-mice. Scale bar: 200 μm. (E) Differences in the density of c-Fos positive cells (cells/mm2) in DG GC layer without and with illumination (medians, interquartile ranges, and individual data). (F) Sketch showing the experimental arrangement: silicon probes inserted into the DG and an optical fiber lowered above the NI to light activate the ChR2-expressing NI GABAergic neurons in vGAT-Cre mice (*n* = 4 mice). Photomicrographs depict the track of the silicone probe in the DG and the position of the optical fiber above the NI. Scale bar: 200 μm for DG and 500 μm for NI images. (G) Sample putative excitatory unit from DG with increasing activity upon light activation of the NI (top). Red dots indicate the average firing rate in the 60-s long periods. Vertical blue shadings denote the laser stimulation periods alternating with baseline periods. PSTH of the same unit (cell) with the corresponding average PSTH (bottom). (H) Summarized data showing all putative excitatory units (*n* = 3) with increasing activity upon NI light stimulation. Vertical blue shadings denote the laser stimulation periods (boxes indicate the median, lower and upper quartiles, red circles depict the individual data). (I) Sample putative inhibitory unit from DG with decreasing activity upon light activation of the NI (top). Blue dots indicate the average firing rate in the 60-s long periods. Vertical blue shadings denote the laser stimulation periods alternating with baseline periods. PSTH of the same unit with the corresponding average PSTH (bottom) (for details, see S3 Extended Data). The data underlying this figure can be found in S1 Data. AAV, adeno-associated virus; ChR2, channelrhodopsin 2; DG, dentate gyrus; GABA, γ-aminobutyric acid; GC, granule cell; NI, nucleus incertus; PSTH, peristimulus time histogram; SOM, somatostatin; vGAT, vesicular GABA transporter.

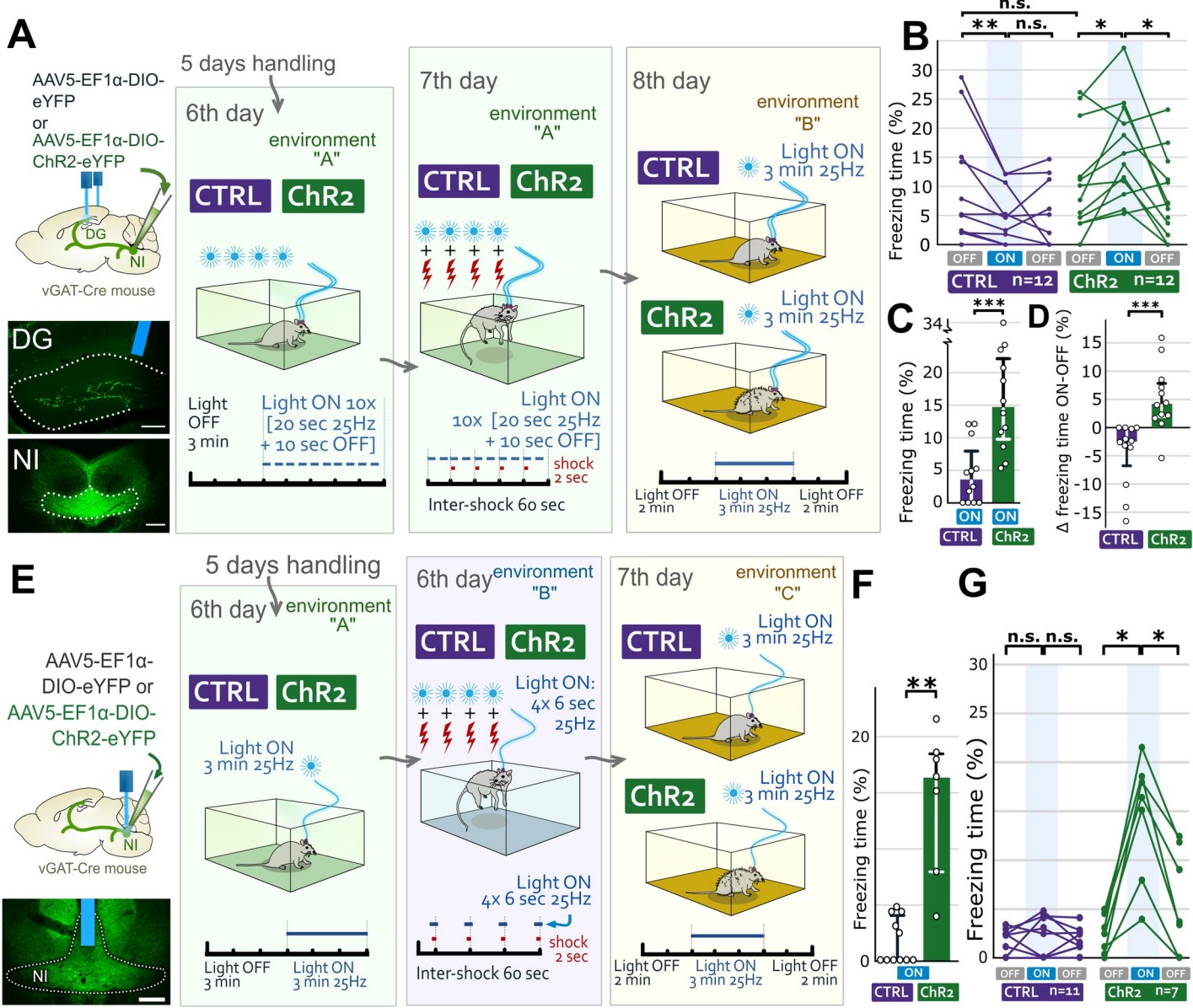

**Fig 4. Reactivation of NI fibers in DG or NI cells activated during memory acquisition recalls fear memory.** (A) Infection of NI GABAergic cells with either control ("CTRL mice") or excitatory opsin-containing AAVs ("ChR2 mice") and bilateral optic fiber implantation over the DG. Representative images: labeling of NI cells and their fibers in DG. Optic fiber is blue. Scale bars: 200 μm. Day 6: baseline freezing behavior of mice without and with light illumination. Day 7: All mice received foot shocks aligned with light illumination. Day 8 in a novel environment B: 2 min OFF 3 min ON 2 min OFF light cycle revealed that optical reactivation of NI fibers could recall fear memory in ChR2 but not in CTRL mice. (B) Individual percentages of time spent with freezing behavior during the light OFF-ON-OFF cycles for each mouse on day 8. (C) Freezing time difference between the 2 groups during the light ON period on day 8 (medians and interquartile ranges). (D) Significant difference in the changes in freezing behavior between the first light OFF and ON periods for both groups (medians and interquartile ranges; for details, see S4 Extended Data). (E) After infecting NI GABAergic cells with excitatory opsin-containing ("ChR2-mice") or control AAVs ("CTRL-mice"), we implanted an optic fiber over the NI. Representative image: injection site and optic fiber position (blue). Scale bar 200 μm. Day 6: baseline freezing behaviors of mice were assessed without and with light illumination. Then, in a novel environment B, all mice received 4 foot shocks precisely aligned with light illuminations. Day 8 in environment C: 2 min OFF 3 min ON 2 min OFF light cycle demonstrated that optical reactivation of NI GABAergic somata could recall fear memory. (F) Freezing time differences between the 2 groups during the light ON period on day 7 (medians and interquartile ranges). (G) Individual percentages of time spent with freezing behavior during the light OFF-ON-OFF cycles for each mouse on day 7 (for details, see S4 Extended Data). The data underlying this figure can be found in S1 Data. AAV, adeno-associated virus; ChR2, channelrhodopsin 2; DG, dentate gyrus; GABA, γ-aminobutyric acid; NI, nucleus incertus; vGAT, vesicular GABA transporter.

of ChR2-mice decreased significantly and became like those of the CTRL-mice (Figs 4B, S5A and S5B). These data demonstrated that reactivation of NI GABAergic fibers, activated in DG during memory acquisition, can recall fear memories, similar to that with the re-inhibition of DG SOM cells.

## Specific activation of NI GABAergic cell bodies can recall fear memory

To confirm that fear memory can be recalled by the direct activation and reactivation of the somata of NI GABAergic cells, similar to that above, we infected the NI GABAergic cells with light-sensitive excitatory opsin-containing ("ChR2-mice") or control AAVs ("CTRL-mice") and implanted an optic fiber above the NI (Fig 4E). After handling, baseline recordings (without and with light) demonstrated no changes in freezing behavior (S5C Fig), indicating that activation of NI GABAergic somata cannot induce fear. The same day, mice received foot shocks precisely associated with the light illumination of a subset of NI GABAergic cells (Fig 4E). The next day, mice were tested using the 2-3-2 min light OFF-ON-OFF paradigm in a novel environment. No difference was found during the first light OFF period (Figs 4G and S5C). Then, illumination of the NI for 3 min (light ON period) induced a significantly increased fear behavior of ChR2-mice (Figs 4F and 4G and S5C and S5D). Behavioral changes were observed immediately in some ChR2-mice, whereas, for the whole group, the increase became significant within 1 min (S5D Fig). After switching off the light, the fear responses of ChR2-mice decreased significantly (Figs 4G and S5C). These findings suggest that activating a subset of NI GABAergic cells can create a specific memory trace during fear memory acquisition, which can be recalled with the reactivation of the same subset of NI GABAergic cells even in a novel environment.

## NI GABAergic cells are activated during fear memory recall in correlation with the activity of DG GCs

To fulfill their role in fear memory recall, NI GABAergic cells need to be activated by salient events during both memory acquisition [34] and recall. To investigate their reactivation, we performed c-Fos immunohistochemistry using vGAT/ZsGreen mice, in which GABAergic cells constitutively express green fluorescent proteins (Fig 5A). After handling and preexposure to environment "A," mice received foot shocks in environment "A" (Fig 5A). The next day, the control mice were sacrificed right after taking them out of their home cages (Home-mice, Fig 5B). Ten mice (Recent-mice), which were returned to environment "A," showed contextual fear behavior (Fig 5E) and were also sacrificed. Then, because the hippocampus is also known to have an active role in remote contextual memory recall [18,46], we exposed 7 mice to the conditioned context 30 days after their conditioning, when they also showed a strong fear response (Remote-mice, Fig 5E). C-Fos immunolabeling revealed that NI GABAergic cells were significantly activated during both recent and remote contextual fear memory recalls and remained relatively silent without a fearful context in their home cages (Fig 5B–5D). DG GCs are also known to remain silent in a home cage and are activated by contextual memory recall [12,47]. Animals likely use different numbers of neurons to encode even similar memory traces; therefore, we tested whether recall-related activation of NI GABAergic cells shows a correlation with DG GC activity. To analyze this, we further investigated c-Fos activity in Recent-mice and Home-mice. First, we found that the number of c-Fos positive GCs was significantly higher in Rec-mice than in Home-mice (S6D Fig). Then, we discovered a significant and strong correlation between the density of naturally activated NI GABAergic cells and naturally activated DG GCs in Recent-mice (Figs 5F, S6A and S6B). However, the correlation was absent in Home-mice, which did not need to recall fear memory within the investigated

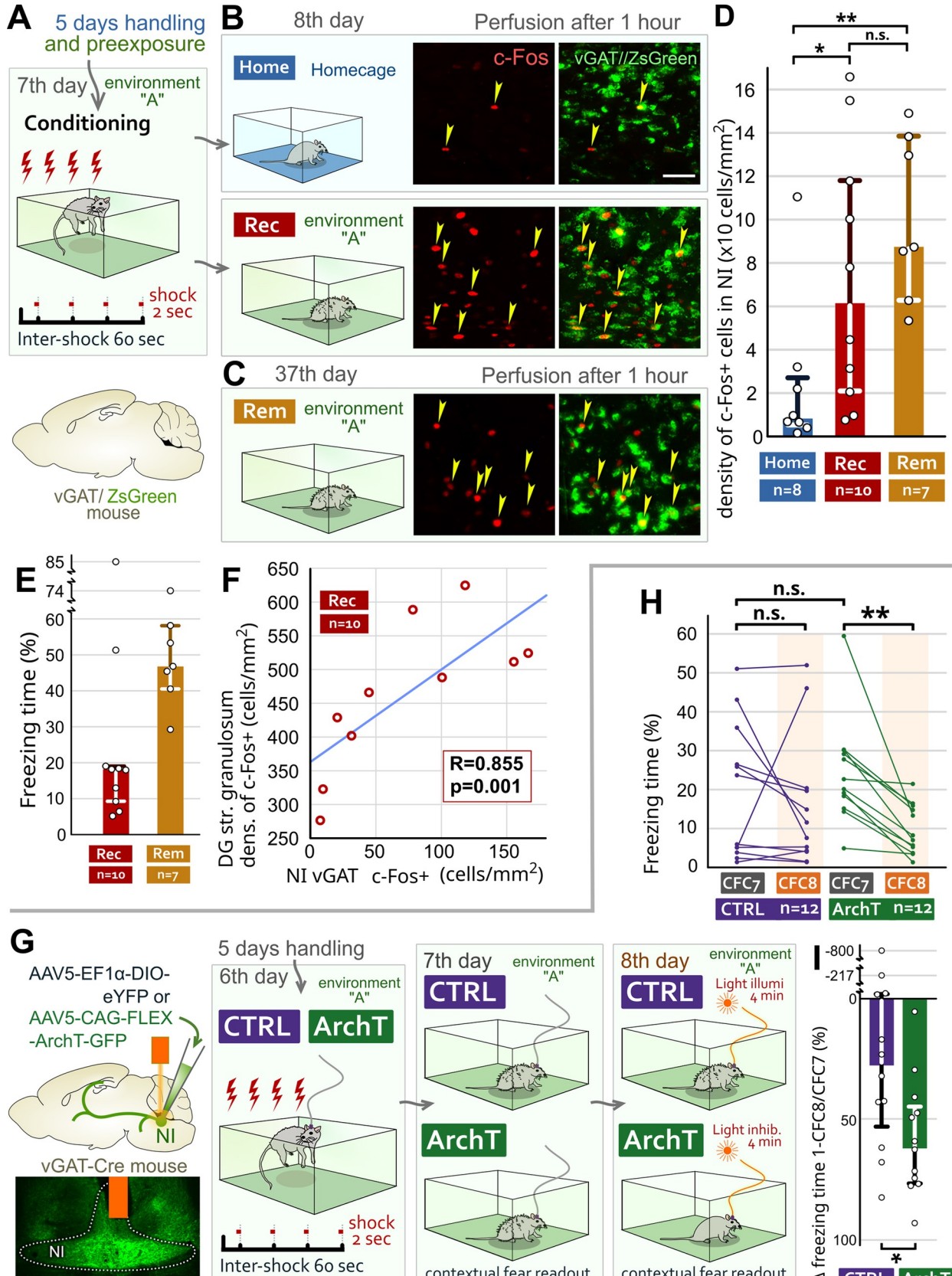

**Fig 5. NI GABAergic cells are activated during contextual fear memory recall, and their activity is necessary for fear memory retrieval.** (A) After handling and preexposure to environment A on day 6, vGAT/ZsGreen mice were placed into environment A on day 7, where they received 4

foot shocks. (B) Day 8: Home-mice were sacrificed immediately after taking them out of their home cages. Recent-mice (Rec) were placed back in environment A for 5 min. Later, we analyzed the c-Fos activity (red) of GABAergic cells (green) in NI. Scale bar: 40 μm. (C) Thirty days later, we tested Remote-mice (Rem) as well. (D) Differences in density (cells/mm2) of c-Fos positive GABAergic cells in NI (medians and interquartile ranges). (E) Contextual fear responses of Recent- and Remote-mice during the 5-min recall period (medians and interquartile ranges). (F) Significant correlation (Spearman-rank correlation) between the density of c-Fos labeling of NI GABAergic cells and DG GCs in Recent-mice. (G) After infecting NI GABAergic cells with ArchT-containing (ArchT mice) or control AAVs (CTRL mice), we implanted an optic fiber over the NI. Representative image: injection site and optic fiber position (orange). Scale bar 200 μm. After handling, mice received 4 foot shocks. On the next 2 days, we analyzed the freezing behavior of these mice without (day 7: "CFC7") and then with (day 8: "CFC8") light illumination of the NI. (H) Individual percentages of time spent with freezing behavior during the light OFF (CFC7) and light ON cycles (CFC8) for each mouse on days 7 and 8, respectively. (I) Significant difference in the changes in freezing behavior between the light OFF (CFC7) and ON (CFC8) periods (medians and interquartile ranges; for details, see S5 Extended Data). The data underlying this figure can be found in S1 Data. AAV, adeno-associated virus; ArchT, archaerhodopsin T-3; DG, dentate gyrus; GABA, γ-aminobutyric acid; GC, granule cell; Home, homecage; NI, nucleus incertus; Rec, recent; Rem, remote; vGAT, vesicular GABA transporter.

period (S6C Fig). These data further suggest that DG GCs are under the disinhibitory control of NI GABAergic cells during contextual fear memory recall.

## NI GABAergic cells are necessary for contextual memory recall

To demonstrate that NI GABAergic cells are required for contextual fear memory recall, we inhibited them (at least partially) during contextual memory recall periods. We infected NI GABAergic cells with light-sensitive inhibitory ArchT-containing ("ArchT-mice") or control AAVs ("CTRL-mice") and implanted an optic fiber above the NI (Fig 5G). After handling, the mice received foot shocks in environment "A" (Fig 5G). The next day, in environment "A," we found no differences in their contextual fear responses without light illumination (CFC7, Fig 5G and 5H). The next day, in environment "A," light illumination significantly decreased the fear behavior of ArchT-mice compared to the previous day, whereas CTRL-mice were unaffected (Fig 5H and 5I). Fear responses were not different between the groups in environment "B" (S6E and S6F). All these results suggest that NI GABAergic cells are necessary for natural contextual fear memory recall.

## Cortical memory centers directly innervate NI neurons that target the hippocampus and can coordinate DG cell activities via NI

Our results indicate that NI GABAergic cells are vital components of the contextual fear memory network. To create the appropriate activity pattern for memory recall, they should receive complex neocortical information. We have previously shown that NI GABAergic cells receive monosynaptic inputs from brain areas that process salient environmental stimuli (including airpuff, auditory tone, available water [34]). Here, we specifically investigated whether hippocampus-targeting NI GABAergic cells receive inputs from key memory processing cortical areas. We labeled hippocampus-targeting NI cells with FluoroGold (FG) from the hippocampus retrogradely, and we labeled principal neurons of the medial prefrontal cortex (mPFC), the ACC, or the retrosplenial cortex (RSC) of vesicular glutamate transporter 1 (vGluT1)-Cre mice (Fig 6A–6G). We found that all 3 cortical areas densely and specifically innervated the NI with characteristically different innervation patterns (Figs 6H–6J, S7A and S7B). They established putative synaptic contacts (identified by Homer-1 postsynaptic protein labeling) on NI neurons that were labeled retrogradely from the hippocampus (Fig 6K–6M). Our anatomical analyses revealed that at least about 80% of hippocampus-projecting NI cells received, on average, about 3 to 4 synapses to their perisomatic regions from these neocortical structures (Fig 6K–6M; see S6 Extended Data). Since only GABAergic cells project from the NI to the hippocampus [34], these data revealed a hitherto unrecognized cortico-incerto-hippocampal disinhibitory pathway.

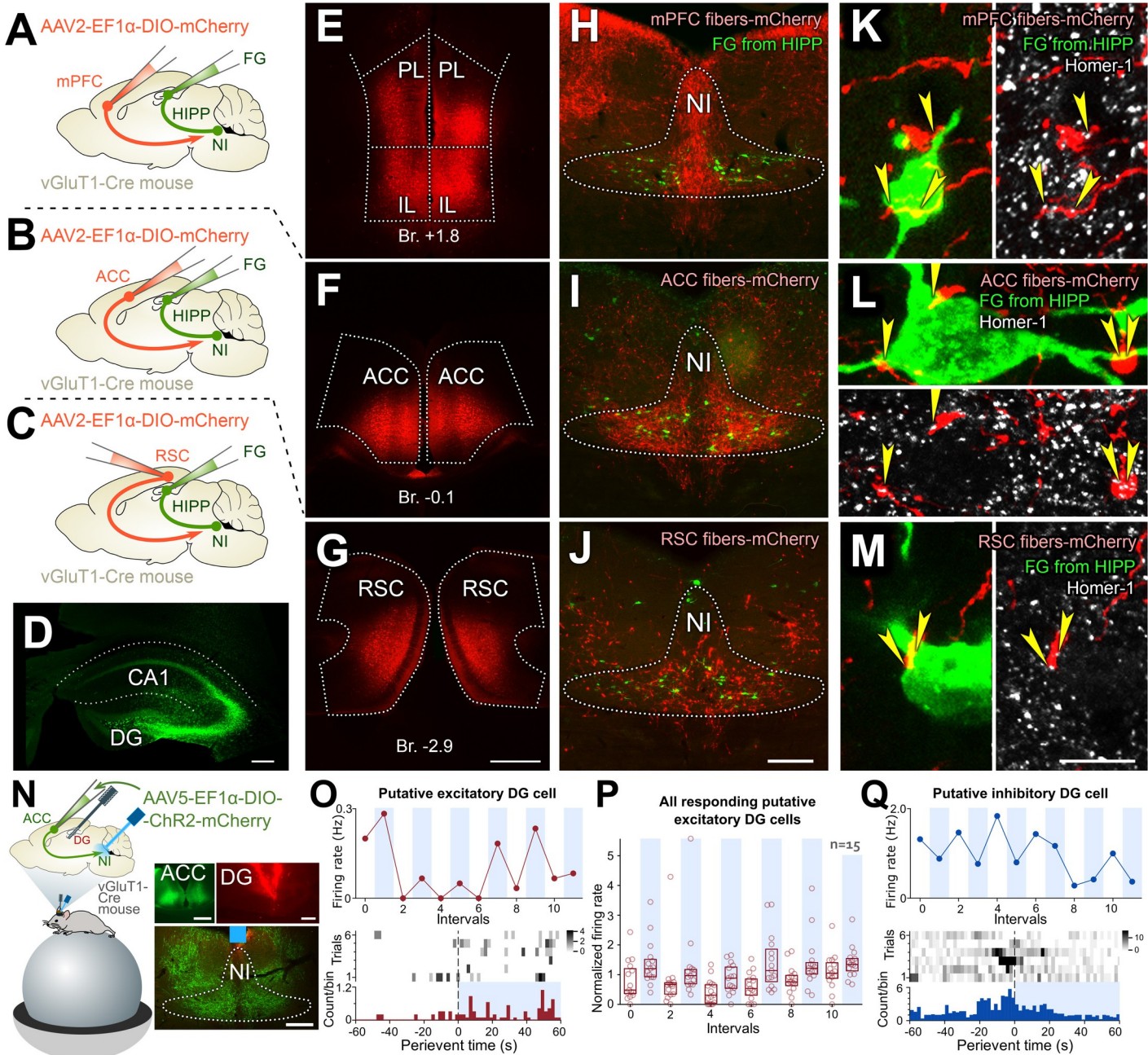

**Fig 6. Cortical memory centers strongly and specifically innervate NI and target hippocampus-projecting NI cells.** (A–C) The anterograde tracer AAV was injected into the mPFC (A, *n* = 2 mice), the ACC (B, *n* = 3 mice), or the RSC (C, *n* = 3 mice) bilaterally, and the retrograde tracer FG was injected into the hippocampus (HIPP) of the vGluT1-Cre mice bilaterally. (D–G) Representative injection sites in the HIPP (D), mPFC (E), ACC (F), and RSC (G). Br.: Bregma position, PL: Prelimbic cortex, IL: Infralimbic cortex. Scale bars: 200 μm for (D), 500 μm for (E–G). (H–J) Innervation pattern (red) of vGluT1-positive cells from the mPFC (H), the ACC (I), or the RSC (J) in the NI (HIPP-projecting retrogradely labeled cells are green). Scale bar: 200 μm. (K–M) Pairs of confocal images show that vGluT1-positive fibers (red) from the mPFC (K), the ACC (L), and the RSC (M) establish Homer-1 labeled (white) synaptic contacts (yellow arrowheads) on FG-positive neurons (green). Scale bar: 10 μm. (N) The experimental arrangement: silicon probes inserted into the DG and an optical fiber lowered above the NI to light activate the ChR2-expressing afferent fibers arriving from the ACC in vGluT1-Cre mice (*n* = 4 mice). Photomicrographs show the injection site in the ACC, the track of the silicone probe in the DG, and the position of the optical fiber above the NI. Scale bars: 200 μm for every images. (O) Sample putative excitatory unit from DG with increasing activity upon optogenetical activation of the ACC afferent fibers in NI (top). Red dots indicate the average firing rate in the 60-s long periods. Vertical blue shadings show the laser stimulation periods alternating with baseline periods. PSTH of the same unit with the corresponding average PSTH (bottom). (P) Summary graph of all putative excitatory units (*n* = 15) with increasing activity after the stimulation of ACC fibers in the NI. Vertical blue shadings show the laser stimulation periods (boxes indicate the median, lower and upper quartiles, red circles depict the individual data). (Q) Sample putative inhibitory unit from DG with decreasing activity upon light activation of the ACC afferent fibers in NI (top). Blue dots indicate the average firing rate in the 60-s long periods. Vertical blue shadings denote the laser stimulation periods alternating with baseline periods. PSTH of the same unit with the corresponding average PSTH (bottom). (For details, see S6

Extended Data). The data underlying this figure can be found in S1 Data. AAV, adeno-associated virus; ACC, anterior cingulate cortex; Br, bregma; ChR2, channelrhodopsin 2; DG, dentate gyrus; FG, FluoroGold; HIPP, hippocampus; mPFC, medial prefrontal cortex; NI, nucleus incertus; RSC, retrosplenial cortex; vGluT1, vesicular glutamate transporter 1.

To explore this functional connectivity in vivo as well along the ACC-NI-DG axis, the ACC was injected with Cre-dependent ChR2-containing viruses bilaterally in vGluT1-Cre mice. In this group of mice, the cortical afferent fibers were light activated in the NI and again multi-channel recordings were performed from the DG (Fig 6N) similar to that above. Although several putative excitatory cells have changed their activity during the light ON periods at least 8.0% (15 from 188 units) of putative excitatory units in DG increased their firing rate consistently and significantly during the laser stimulation periods (Fig 6O and 6P) and at least 1.4% (2 from 140) of putative inhibitory DG units decreased their firing rate upon laser stimulation (Fig 6Q). Although these indirect optical stimulations may have a very low efficiency, this ratio is very close to the known ratio of the cells that participate in any given event-related cell assembly in the DG and demonstrate that even a higher order cortical center (ACC) can influence DG cell activities via the brainstem NI—DG SOM cell pathway.

## Discussion

Contextual memory formation has been under investigation for decades. DG GCs play a key role in this process [7–9,12,14]. They receive context-related multimodal sensory information from the entorhinal cortex that DG can convert to contextual codes [8,15,17,48]. In the last decade, several seminal papers and reviews have demonstrated that these codes are maintained by a small subset of DG GCs that create a memory engram that serves as an index of memory content [1,2,4–6,11–13]. The reactivation of engram cells is both sufficient and necessary to recall the encoded contextual memory [9–13]. In the DG and amygdala, the effective allocation of principal cells to a memory engram correlates with the excitability of the principal cells [10,19,20,49]. Hippocampal dendrite-targeting SOM interneurons are effective regulators of this excitability [8,16,31,32] and SOM interneurons have an important role in the formation, consolidation, and recall of a memory traces across several brain regions [16,22,32,50]. Therefore, we hypothesized that they could also control the selection of engram cells in the hippocampus. Here, using c-Fos experiments, we found that inhibiting DG SOM cells activates GCs by disinhibition. This can also explain our other findings that the precise re-inhibition of the same DG SOM cells, which were inhibited during memory acquisition, can recall fear memories (encoded in the DG during contextual conditioning). This was specific because if the experimental inhibition of DG SOM cells was not associated with a fearful event, then their re-inhibition could not recall any fear memories in a novel environment. Although DG plays a key role in contextual memory recall [7–9,12,14], the extent of which the DG also recalled the contextual elements of the frightening events cannot be tested. Although the DG likely recalled the conditioned context that frightened the mice, but it cannot be excluded that the hippocampus also has more direct pathways to recall encoded fear memories. In addition, chemogenetic, temporally, and spatially nonspecific inhibition of DG SOM cells did not enhance contextual fear recall but did significantly disrupt it, because this kind of inhibition could overwrite the original inhibitory pattern of DG SOM cells. These findings indicate that the unspecific disinhibition of a GC population cannot recall fear memories even if that population partially overlaps with the widely distributed population of GCs that encodes a strong and recent fear memory. Thus, the pattern of hippocampal SOM cell inactivity needs to be temporally and spatially precise for memory recall. We also demonstrated that DG SOM cells are both sufficient and necessary for natural fear memory recall since nonspecific inhibition of a subpopulation of DG SOM

cells disrupted contextual fear recall in the fear-conditioned context. Furthermore, inhibition of PV cells cannot allocate more GCs to an engram [16], which may explain why recall was not possible by inhibiting DG PV interneurons, even if they were inhibited during fear conditioning as well, which further suggested a specific role for the SOM cells.

Hippocampal CA1 pyramidal cells receive contextual information from DG (via CA3 neurons) and less processed sensory inputs from the entorhinal cortex simultaneously during fear learning [28,32,51]. Thus, CA1 pyramidal cells can encode associations or the experience of the events rather than only their context [2,52]. Inhibition of CA1 SOM cells only during contextual memory acquisition prevented fear memory recall in a previous study [32], whereas it can have a different effect in the DG [16]. Here, we demonstrated that in that previous study, a fear memory was actually created but remained silent because of the lack of a proper recall pattern from CA1 SOM cells. However, if we re-inhibited the same CA1 SOM cells that were inhibited during the fearful events, we could recall the memory.

Entorhinal inputs alone do not seem to be suitable for memory recall because HIPP CA1 PC and DG GC encode information about location and context more efficiently and accurately than their entorhinal inputs alone [15,53]. This suggested that extra-cortical inputs must contribute to the refinement of the representation of these memory traces, one of which input is presented here. Hippocampal fibers of NI GABAergic cells are readily activated by salient contextual stimuli in vivo [34,40]. DG and CA1 hippocampal areas encode different aspects of memory and SOM interneurons of both areas are specifically innervated by NI GABAergic cells. Therefore, NI GABAergic cells seem ideal both supporting and coordinating memory formation in different hippocampal areas. Studies have already indicated that inhibition of NI can disrupt spatial navigation and spatial memory recall [41,42]. Previously, we also showed that activating NI GABAergic cells during contextual memory acquisition prevented fear memory recall if they were not re-activated during the memory recall period [34]. However, here, we demonstrated that the activation of a subset of NI GABAergic cells during fear memory acquisition could associate them with a fear memory trace. This memory trace may remain silent, but the presentation of the same activity pattern of NI GABAergic cells or their DG-projecting fibers can recall fear memory. In addition, NI activity is not only sufficient but also necessary for natural memory recall because inhibition of NI GABAergic cells during the memory retrieval period disrupts fear memory recall. Furthermore, our c-Fos experiments supported our conclusions because NI GABAergic cells were highly active during contextual fear memory recall, and their activity was positively correlated with that of DG GCs during memory recall. In contrast, no such correlation was detected in the home cage. Moreover, our in vivo electrophysiological recordings from DG during NI stimulation showed that NI could reorganize network activity, inhibit putative interneurons, and disinhibit putative excitatory cells in the DG. This further confirms that NI GABAergic cells can disinhibit DG GCs via the SOM cells to encode and recall fear memories.

Accurate memory recall needs to be coordinated by neocortical centers, such as the mPFC, ACC, or RSC, that can initiate memory recall by re-activating engram cells in the hippocampus [18,54–57]. Our results demonstrated that glutamatergic fibers of the mPFC, ACC, and RSC give abundant and specific innervation to the NI and target the NI GABAergic cells that project to the hippocampus. Moreover, our in vivo electrophysiological recordings show that stimulation of ACC fibers in NI could modulate DG cell activities in a ratio similar to that estimated during engram formation, suggesting functional connection between ACC and DG via the pontine NI. This previously unrecognized direct cortico-incerto-hippocampal pathway likely creates an essential disinhibition-based mechanism for selecting and re-activating hippocampal engrams during memory processing (for a summary, see S10 Fig).

Understanding the role of this direct cortico-incerto-hippocampal pathway could have far-reaching clinical consequences. NI GABAergic/relaxinergic fibers abundantly arborize in the

primate hippocampus [58], and a recent study demonstrated that the human brainstem region containing NI is specifically activated during memory recall in object recognition tests [59]. Our findings suggest that the dysfunction of hippocampal SOM cells in Alzheimer's disease [60,61] may have a key role in the memory impairment associated with this disease. Dementias and amnesias are often associated with silent engrams that cannot be recalled by natural stimuli [5,14,62,63]; however, neocortex disinhibition through, for example, reduced GABA levels can aid memory recall in humans [30]. Our findings may also explain how the loss of hippocampal SOM cells can contribute to the cognitive symptoms of schizophrenia and why it may also impair pattern separation in DG [64–66]. Understanding vulnerabilities in this cortico-incerto-hippocampal pathway may be vital for finding future treatments for these diseases.

## Materials and methods

### Ethics statement

All experiments were performed in accordance with the Institutional Ethical Codex and the Hungarian regulations on animal research (Act XXVIII of 1998, Government Decree 40/2013), in line with the EU Directive 2010/63/EU. The Animal-welfare Body of the Institute of Experimental Medicine and the Government Office of Pest County authorized the experiments at project number PE/EA/2553-6/2016.

### Mice

We used PV-iRES-Cre, vGAT-iRES-Cre, vGluT1-iRES-Cre, vGAT-iRES-flpo, Gt(ROSA)26Sor-CAG/LSL-ZsGreen1 mice (all from The Jackson Laboratory), heterozygous SOM-iRES-Cre mice (courtesy of Prof Josh Huang), and wild-type C57Bl/6 mice (from Charles River). We also crossbred mice to have vGAT-iRES-flpo/SOM-iRES-Cre and vGAT-iRES-Cre/Gt(ROSA)26Sor-CAG/LSL-ZsGreen1 mice. We used adult (at least 6 weeks old) mice from both sexes in our experiments. Mice had access to food and water ad libitum and were housed in a vivarium (3 to 5 mice/cage) until used in experiments. Mice used for optogenetic and chemogenetic behavioral experiments were maintained on a normal 12-h light-dark cycle, with experiments performed during the light phase of the cycle.

### Viruses

The viruses used in this study are subject to a material transfer agreement (MTA). AAV2/5-EF1α-DIO-eYFP, AAV2/5-EF1α-DIO-ChR2(H134R)-eYFP, AAV2/2-EF1α-DIO-mCherry, AAV2/5-EF1α-DIO-mCherry, AAV2/5-EF1α-DIO-ChR2(H134R)-mCherry, and AAV2/5-CAG-FLEX-ArchT-GFP were obtained under an MTA with UNC Vector Core. AAV2/1-hSyn-SIO-eOPN3-mScarlet was obtained under MTA with O. Yizhar at the Weizmann Institute of Science. AAV2/1-hSyn-SIO-eOPN3-mScarlet and AAV2/8-hSyn-DIO-hM4D(Gi)-mCherry were obtained under MTA with Addgene. Virus AAV2/1-EF1α-fDIO-eYFP, AAV2/5-EF1α-FLEX-TVA-mCherry, AAV2/5-CAG-FLEX-oG were obtained under MTA with the Salk GT3 Vector Core. Rabies(ΔG)-EnvA-GFP was obtained under MTA with Charité-Universitatsmedizine Berlin.

### Stereotaxic surgeries for viral gene transfer, retrograde tracing, and optic fiber implantations

Mice were anesthetized with 2% isoflurane followed by an intraperitoneal injection of an anesthetic mixture (containing 8.3 mg/ml ketamine and 1.7 mg/ml xylazine-hydrochloride in 0.9% saline, 10 ml/kg body weight) and were then mounted in a small animal stereotaxic frame

(David Kopf Instruments, California, United States of America) and the skull surface was exposed. A Nanoject II precision microinjector pump (Drummond, Broomall, Pennsylvania, USA) was used for the microinjections. For anterograde tracing, optogenetic and chemogenetic experiments, we injected 30 to 100 nl of one of the following viruses into the target brain areas: AAV2/5-EF1α-DIO-eYFP; AAV2/2-EF1α-DIO-mCherry; AAV2/5-EF1α-DIO-mCherry; AAV2/5-EF1α-DIO- hChR2(H134R)-mCherry; AAV2/5-EF1α-DIO-hChR2 (H134R)-eYFP; AAV2/5-CAG-FLEX-ArchT-GFP (the viruses above were from UNC Vector Core), AAV2/1-EF1α-fDIO-eYFP (the virus was from Salk GT3 Vector Core), AAV2/1-hSyn-SIO-eOPN3-mScarlet and AAV2/8-hSyn-DIO-hM4D(Gi)-mCherry (the last 2 viruses were from Addgene; $4.4$–$21 \times 10^{11}$ colony forming units/ml for all viruses). We always used AAV2/1 viruses in concentration lower than $10^{12}$ to avoid any anterograde trans-synaptic labeling [67], and indeed we have never observed such labeling. For retrograde tracing experiments, we injected 100 nl of 2% FluoroGold (Fluorochrome, Denver, Colorado, USA) into the target areas. The coordinates for the injections were defined by a stereotaxic atlas; the null coronal plane of the anteroposterior (AP) axis was defined by the position of Bregma; the null sagittal plane of the mediolateral (ML) axis was defined by the sagittal suture; the null horizontal plane of the dorso-ventral (DV) axis was defined by the positions of Bregma and Lambda points. The injection coordinates were the following (always given in mm at the AP, ML, and DV axes, respectively): dentate gyrus: −2.2, +/−1.1, −2.2 (1–1 injections bilaterally), hippocampus CA1: −2.2, +/−1.4, −1.4 (1–1 injections bilaterally), nucleus incertus: −5.0, 0.0, −4.3, medial prefrontal cortex: +1.8, +/−0.3, −3.0 (1–1 injections bilaterally), anterior cingulate cortex: −0.3, +/−0.3, −1.5 (1–1 injections bilaterally), retrosplenial cortex: −2.8, +/−0.3, −1.1 (1–1 injections bilaterally), hippocampus for retrograde experiments: −2.0, +/−1.5, −2.0 and −3.3, +/−3.0, −3.0 (2–2 injections bilaterally). After the surgeries, mice received 0.3 to 0.5 ml saline and 0.1 mg/kg meloxicam (Metacam, Boehringer Ingelheim, Germany) intraperitoneally and were placed into separate cages until further experiments or perfusions.

## Optic fiber implantations for behavioral and optogenetic experiments with c-Fos staining

For behavioral and optogenetic experiments with c-Fos staining, virus injections were followed by optic fibers surgeries. These were similar to the virus injections and were carried out 5 to 6 weeks after virus injections. Optic fibers (105 μm core diameter, 0.22 NA, Thorlabs GmbH, Dachau/Munich, Germany) were implanted into the brain with the tip at the following coordinates: nucleus incertus: −5.0, 0.0, −4.1; dentate gyrus: −2.2, +/−1.0, −1.8; hippocampus CA1 region: −2.2, +/−1.4, −0.9. For secure fixture of the implantable optic fiber, 3 screws were inserted into the skull followed by disinfection and drying the surface with 70% ethanol and finally dental cement (Paladur, M+W Dental, Hungary) was added between the skull and the base of the ceramic ferrule of the fiber implant (Precision Fiber Products, California, USA). For behavioral experiments to identify the positions of the optic fibers, the tips of the fibers were labeled with DyI or DyO (Thermo Fischer, USA) for animals with eYFP- or mCherry/mScarlet-expressing viruses, respectively. Positions of the optic fibers are illustrated in S8 and S9 Figs. After the surgeries, mice received 0.3 to 0.5 ml saline and 0.1 mg/kg meloxicam (Metacam, Boehringer Ingelheim, Germany) intraperitoneally and were placed into separate cages until experiments or perfusions.

## Surgical procedure for in vivo electrophysiology experiments

Approximately 5 to 6 weeks after virus injections, surgeries were performed under general anesthesia (isoflurane 0.5% to 1.5%). A small lightweight headplate was attached to the skull

using Paladur dental acrylic (Kulzer, Hanau, Germany). A cranial window (1.8 mm diameter) above the right hippocampal formation (AP, −2.1 mm; ML, −1.3 mm) and a hole above the NI (AP, −6.5 mm; ML, 0 mm) was drilled under stereotaxic guidance. For ground electrode, a hole was drilled above the cerebellum (AP, −5.6 mm; ML, 2.0 mm left side). The craniotomy and the holes were covered with fast sealant (Body Double, Smooth-On, Easton, Pennsylvania, USA). After surgery, the mice were continuously monitored until recovered, then they were returned to their home cages.

## Mono-trans-synaptic rabies tracing

SOM-Cre mice were prepared for stereotaxic surgeries as described above, and 60–60 nl (into the dorsal DG, −2.2, +/−1.1, −2.2), and 100–100 nl (into the ventral DG, −3.6, +/−2.6, −3.0) of the virus combination AAV2/5-EF1α-FLEX-TVA-mCherry + AAV2/5-CAG-FLEX-oG (diluted in 1:1, Salk GT3 Core, $4.5 \times 10^{12}$ colony forming units/ml) were injected into DG bilaterally. These Cre-dependent viruses contain an avian tumor virus receptor A (TVA), which is necessary for them to be infected by the rabies viruses and they contain an upgraded version of the rabies glycoprotein (oG) that provides an increased trans-synaptic labeling potential for the rabies viruses [68]. After 4 weeks of survival, mice were injected with the genetically modified Rabies(ΔG)-EnvA-GFP (Charité Uni.) 50-50-50-50 nl at the same coordinates. After 7 days of survival, mice were prepared for perfusions. After immunohistochemistry, we visualized every brain area where we could detect rabies-labeled input cells in both mice.

## Perfusions

Mice were anesthetized with 2% isoflurane vapor followed by an intraperitoneal injection of an anesthetic mixture (containing 8.3 mg/ml ketamine, 1.7 mg/ml xylazine-hydrochloride, 0.8 mg/ml promethazinium-chloride) to achieve deep anesthesia. Mice were then perfused transcardially with 0.1 M phosphate-buffered saline (PBS, pH 7.4) solution for 2 min, followed by 4% freshly depolymerized paraformaldehyde (PFA) solution for 45 min, followed by PBS for 10 min, then the brains were removed from the skull. After perfusions, brains were cut into 50 to 60-μm thick sections using a vibrating microtome (Leica VT1200S or Vibratome 3000).

## Antibodies

The list and specifications of the primary and secondary antibodies used can be found in S1 and S2 Tables. The specificities of the primary antibodies were extensively tested, using either knock-out mice or other reliable methods. Secondary antibodies were extensively tested for possible cross-reactivity with the other antibodies used, and possible tissue labeling without primary antibodies was also tested to exclude auto-fluorescence or specific background labeling. No specific-like staining was observed under these control conditions. Combinations of the used primary and secondary antibodies in the different experiments are listed in S3 Table.

## Fluorescent immunohistochemistry and laser-scanning confocal microscopy for counting cells and synaptic contacts

Perfusion-fixed sections were washed in 0.1 M PB (pH 7.4) and incubated in 30% sucrose overnight for cryoprotection. Sections were then freeze-thawed over liquid nitrogen 3 times for antigen retrieval. Sections were subsequently washed in PB and Tris-buffered saline (TBS, pH 7.4) and blocked in 1% human serum albumin in TBS (HSA; Sigma-Aldrich) and then

incubated in a mixture of primary antibodies for 48 to 72 h. This was followed by extensive washes in TBS and incubation in the mixture of appropriate secondary antibodies overnight. We used 4′,6-diaminido-2-phenylindole (DAPI) staining (Sigma-Aldrich) to visualize cell nuclei. Then, sections were washed in TBS and PB, put on slides and covered with Aquamount (BDH Chemicals). For the behavioral experiments, viral anterograde, and retrograde tracing experiments, each injection site was reconstructed from 50 to 60 μm sections using a Zeiss Axioplan2 microscope. Every part of the injected tissue containing even low levels of tracer was considered as part of the injection site. For anatomical analysis, sections were evaluated using a Nikon A1R confocal laser-scanning microscope system built on a Ti-E inverted microscope with a 10× air objective or with a 0.45 NA CFI Super Plan Fluor ELWD 20XC or with a 1.4 NA CFI Plan Apo VC 60× oil objective both operated by NIS-Elements AR 4.3 software. Regions of interest were reconstructed in z-stacks; distance between the focal planes was 0.5 μm for examined synaptic contacts and 2 to 3 μm for examined neuronal somata. The cell counting was performed using the NIS-Elements AR 4.3 or Adobe Photoshop CS6 Extended software. Finally, we estimated that in optogenetic experiments, we illuminated about 0.1% ($0.027 \text{ mm}^3 / 25.7 \text{ mm}^3$) of the whole mouse hippocampus, where about 55% of SOM cells were infected by viruses, whereas, in chemogenetic experiments, we inhibited about 40% of hippocampal DG SOM cells.

## Analysis of the cortical innervation of NI

To quantify differences in innervation pattern of NI by cortical areas, we used immunohistochemistry and Zeiss Axioplan2 microscope for epifluorescent imaging. Every third sections of the NI were reconstructed (about 3 sections of NI per animal from Bregma −5.30 to −5.60) and analyzed. We used Adobe Photoshop CS6 Extended software. First, we defined the border of the NI according to the mouse anatomical atlas, then we divided its area into 10 equally wide sub-areas in medio-lateral range. We measured and normalized the median pixel intensities of every NI sub-area innervated by different cortical inputs. The results are demonstrated in S7 Fig.

## Optogenetic experiments for counting c-Fos labeled cells

After optic fiber implantations (without DyI or DyO), mice received 5 days of handling. On the sixth day, mice were placed into a chamber (40 cm × 20 cm × 20 cm, divided into 2 areas with striped walls and floor on the one side and dotty walls and floor on the other side) that was enriched with a specific combination of olfactory (macadamia nut scent), visual (striped and dotted walls and floors), and auditory (white noise) cues and novel objects (paper roll) for a complex contextual experience. Mice were allowed to move freely for 10 min in this context, where eOPN3-mice received 5 s of light illuminations 20 times (10 mW intensity at the tip of the optic fiber at 473 nm wavelength with 25 s inter-light interval). Whereas NI fiber stimulated CTRL and ChR2 mice received a 10-min long light stimulation (15 ms pulses at 25 Hz with 10 mW intensity, 473 nm). Laser illuminations were started immediately before we placed mice into the context. In these experiments, we always illuminated only one of the hemispheres. The experiments were performed in a counterbalanced way, some mice received light illumination above the right DG, while others above the left DG. After the spatial exploration, mice were placed back into their home cages and after 40 to 60 min they were sacrificed to label c-Fos-positive cells in the DG activated by the paradigms described above. Every section under the optic fibers in a 300 μm thickness range (either on the light illuminated or on the non-light illuminated side) was evaluated using Zeiss Axioplan2 microscope. The cell counting was performed using the Adobe Photoshop CS6 software, and investigators were blinded to

the treatment of the hemispheres at all stages of the analyses until all counting was completed. We counted c-Fos-positive cells only in the stratum granulosum of the DG, the borders of which were defined by DAPI labeling of the nuclei of GCs. The number of the counted cells was normalized to the size of the DG GC layer measured by FIJI ImageJ software.

## Imaging and analysis of c-fos experiments in the NI and in the hippocampus in vGAT/ZsGreen mice

After the immunohistochemistry procedure, we used a Pannoramic Midi II automatic slide scanner (3DHISTECH) equipped with a pco.edge 4.2 camera (PCO) for widefield epifluorescent imaging. The NI was reconstructed in z-stacks (2 μm steps) on every second section (4 NI sections per animal, from Bregma −5.30 to −5.60) using a Zeiss Plan-Apochromat 20×/0.8 objective. In case of the hippocampus, we used a Zeiss Plan-Apochromat 10×/0.3 objective to reconstruct every sixth section (2 × 5 per animal) of the dorsal hippocampus (from Bregma −1.30 to −2.60) in z-stacks (5 μm steps). The border of the NI was defined by the mouse anatomical atlas, whereas the border of the stratum granulosum of the DG was defined by the dense DAPI staining of the layer. After creating 2D focus stacked images, we performed automatic cell counting using NIS-Elements 5.3 software (Nikon, GA3 Module). Following background subtraction on each section, we used the Bright Spots detection tool to count c-Fos immunopositive cells with mean pixel intensity greater than 15,000 (image bit depth is 16-bit). Cell numbers were divided by the respective area (NI or stratum granulosum of the DG) to gain comparable cell densities (1/mm2).

## In vivo multichannel electrophysiological recordings

During multichannel recordings, mice were head-restrained with a downward tilted head position (pitch angle: 20o). Probes and optical fiber were coated with DiI (Thermo Fisher Scientific, Waltham, Massachusetts, USA) for later histological verification of the location. UCLA128 channel silicon probes [69] (Masmanidis lab, UCLA 128K, Los Angeles, California, USA) were lowered through the cranial window above the right dorsal DG under isoflurane anesthesia (0,75–1,5%). An optical fiber (200 μm core diameter, 0.22 NA, Thorlabs) was inserted above the nucleus incertus (AP, −6.5 mm; ML, 0 mm, DV, 4.4 mm, in a 20o angle). Ground electrode was placed above the cerebellum. Mice were allowed to recover from anesthesia for approximately 1 h before further lowering the silicon probes. Head restrained mice were free to run, walk, or sit on an air supported, free-floating, 20 cm diameter polystyrene ball. The probes in the hippocampal formation were advanced using a micromanipulator (Luigs & Neumann, Ratingen, Germany) until reaching the GC layer, identified by increased occurrence of unit activity and the appearance of dentate spikes (S4B Fig). Recording was commenced after an approximately 1 h waiting period for letting the tissue settle around the probes. Electrophysiological recordings were performed by a signal multiplexing head-stage (RHD 128, Intan Technologies, Los Angeles, California, USA) and an Open Ephys data acquisition board (open-ephys.org). Signals were acquired at 20 kHz sample rate (data acquisition software: Open Ephys v0.6.3). Following a 10-min long control recording, laser stimulation (60 s 25 Hz, pulse length 5 ms, 10 to 15 mW intensity) and baseline periods (60 s) were alternated 6 to 10 times. Custom-built microprocessor-based (Arduino) behavioral control system enabled the delivery of laser stimulation. At the end of the recording, mice were transcardially perfused with 4% paraformaldehyde, and the brain was removed for post hoc immunohistochemistry. All in vivo data were analyzed in Igor Pro 9 (Wavemetrics, Lake Oswego, Oregon, USA).

## Spike sorting and neuron classification for in vivo electrophysiology experiments

Neuronal spikes were detected and automatically sorted from the high pass filtered (0.3 to 6 kHz) recordings by a template matching algorithm using the Spyking Circus software [70], followed by manual curation of the clusters using the Phy software [71] to obtain well-isolated single units. Multiunit or noise clusters were discarded from the analysis. Spike sorting quality was assessed with refractory period violation and visual inspection of auto- and cross correlations; poor quality clusters were discarded. A burst index was computed by calculating the ratio of the average values in 2 to 10 ms and 10 to 100 ms windows of the single units' autocorrelograms. Putative inhibitory neurons were defined if the burst index was less than 4, while putative excitatory neurons were defined if the burst index was more than 4 [72] (S4C Fig). Units were defined as responsive to the laser stimulation if their firing rate significantly deviated from the baseline activity upon laser stimulation (Wilcoxon signed-rank test $p < 0.05$). In Figs 3H and 6P, unit firing rates were normalized by their average values.

## Optogenetic re-inhibition of DG SOM, PV, and CA1 SOM cells

After optic fiber implantations, mice were transferred to an animal room in the behavioral unit of the institute to rest, then they received 5 days of handling. On the sixth day (baseline day), mice were placed into the first environmental context (environment "A," 20 cm × 20 cm × 20 cm chamber, with striped walls and striped floor, washed with macadamia nut scent) and were allowed to freely move for 3 min to record baseline freezing levels (baseline light OFF period). Then, mice received a 3-min long laser light illumination (10 mW intensity at the tip of the optic fiber at 473 nm wavelength for eOPN3 or 589 nm wavelength for ArchT opsins) to record baseline freezing levels during light illumination (baseline light ON period). Mice displaying higher than 5% baseline freezing levels in environment "A" during baseline periods were excluded from further experiments. On the seventh day (association day), mice were placed into the second environmental context (environment "B"), into a plexiglass foot-shocking chamber (25 cm × 25 cm × 30 cm) that was enriched with a specific combination of olfactory ("baby soap" scent), visual (dim red room lighting), and tactile (metal bars on the floor) cues. Mice were allowed to freely move in the second environment for 1 min. After this, mice received 4 foot shocks (2 s, 2 mA intensity, 58 s inter-shock interval). For "CTRL-mice" and "Associated eOPN3-mice," foot shocks were paired with 6-s long blue laser light illumination (10 mW intensity at the tip of the optic fiber at 473 nm wavelength) that was precisely aligned with the shocks, starting 2 s before the shock onset and finishing 2 s after shock offset. "Not associated eOPN3-mice" did not receive light illumination on day 7. For CA1 experiments, CTRL and ArchT-mice received foot shocks that were paired with 6-s long yellow laser light illumination (10 mW intensity at the tip of the optic fiber at 593 nm wavelength) that was precisely aligned with the shocks. All mice displayed equally strong immediate reactions to foot shocks. After receiving the last shock, mice were kept in the context for another 1 min. After 4 successfully delivered shocks, mice were placed back into their home cages for 24 h. On the eighth day (readout day), mice were placed into the third environmental context (environment "C," 40 cm × 40 cm × 60 cm chamber) with distinct olfactory (argan oil scent), visual (brown curved walls), and tactile (gray plastic floor) cues. Initially, as expected, mice showed some freezing behavior because it was not possible to build an experimental paradigm that is completely different from environment "B" [73]. Even if tactile, visual, and olfactory cues are different, the way the researcher handles mice during the experiments, the way they are placed into a plastic case without bedding and some of the noises are inevitably similar. However, after the mice realized that they are in a different environment, this fear quickly decreased to a

baseline level during this initial period (S1A Fig) and it did not differ among the groups (Figs 1B and S1C). Mice were allowed to freely move in environment "C" for 5 min, where the last 2 min were recorded as the first OFF period (readout OFF period). Then, mice received a 3-min long laser light illumination (10 mW intensity at the tip of the optic fiber at 473 nm wavelength for eOPN3 or 593 nm wavelength for ArchT opsins), recorded as readout ON period. After the termination of the light illumination, mice were kept in the environment "C" for 2 min to read out post-light freezing levels (a second readout OFF period). On the ninth day (contextual readout day), 5 "Associated eOPN3-mice" and 8 "Not associated eOPN3-mice" were replaced into environment "B" for 3 min to detect their contextual fear behavior.

## Aligned and shifted inhibition of DG SOM cells

After optic fiber implantations, mice were transferred to an animal room in the behavioral unit of the institute to rest, then they received 5 days of handling. On the sixth day (association day), mice were placed into the first environmental context (environment "A") into a plexi-glass foot-shocking chamber (25 cm × 25 cm × 30 cm) that was enriched with a specific combination of olfactory ("macadamia soap" scent), visual (dim red room lighting, striped walls), and tactile (metal bars on the floor) cues. Mice were allowed to freely move for 3 min. After this, mice received 4 mild foot shocks (2 s, 2 mA intensity, 118 s inter-shock interval). For "Aligned-mice," foot shocks were paired with 6-s long yellow laser light illumination (10 mW intensity at the tip of the optic fiber at 593 nm wavelength), which was precisely aligned with the shocks, starting 2 s before the shock onset and finishing 2 s after shock offset. "Shifted-mice" received 4 × 6 s long yellow laser illumination 58 s after each foot shock. After receiving the last shock, mice were kept in the context for another 1 min. After 4 successfully delivered shocks, mice were placed back into their home cages for 24 h. On the seventh day (readout day), mice were placed into the second environmental context (environment "B," 40 cm × 40 cm × 60 cm chamber) with distinct olfactory (baby soap scent), visual (gray curved walls), and tactile (gray plastic floor) cues. Mice were allowed to freely move in environment "B" for 6 min, where the last 2 min were recorded as the first OFF period (readout OFF period). Then, mice received a 3-min long laser light illumination (10 mW intensity at the tip of the optic fiber at 593 nm wavelength), recorded as readout ON period. After the termination of the light illumination, mice were kept in environment "C" for 2 min to read out post-light freezing levels (a second readout OFF period). On the eighth day (contextual readout day), mice were replaced into environment "A" for 6 min, where the last 2 min were recorded as the first OFF period (readout OFF period). Then, mice received a 3-min long laser light illumination (10 mW intensity at the tip of the optic fiber at 593 nm wavelength), recorded as readout ON period. After the termination of the light illumination, mice were kept in environment "A" for 2 min to read out post-light freezing levels (a second readout OFF period).

## Chemogenetic inhibition of DG SOM cells during contextual fear recall

Five weeks after the virus was injected, mice were transferred to the animal room of the behavioral unit of the institute to rest, then they received 5 days of handling. On the sixth day, mice were placed into environment "A" that was a plexiglass foot-shocking chamber (25 cm × 25 cm × 30 cm) that was enriched with a specific combination of olfactory ("baby soap" scent), visual (striped walls), and tactile (metal bars on the floor) cues. Mice were allowed to freely move for 3 min to record baseline freezing levels. Mice displaying higher than 5% baseline freezing levels were excluded from further analysis. Then, mice received 4 foot shocks (2 s, 2 mA intensity, 58 s inter-shock interval). All mice displayed equally strong immediate reactions to foot shocks. After receiving the last shock, mice were kept in the context for another 1 min.

After 4 successfully delivered shocks, mice were placed back into their home cages for 24 h. On the seventh day, both "hM4Di-mice" (which had a clozapine-N-oxide (CNO)-sensitive inhibitory G-protein-coupled receptor) and the CTRL (control)-mice were injected intraperitoneally with CNO (3 mg/kg in 10 ml, Tocris). An hour after the injection, mice were placed back to environment "A," where we analyzed the first 3 min of their contextual fear behavior. After behavior experiment, we performed SOM immunohistochemistry on every sixth dorsal hippocampus-containing slices (from Bregma −1.30 to −2.60) for cell counting.

## Optogenetic re-stimulation of NI fibers in DG

After optic fiber implantations, mice were transferred to the animal room of the behavioral unit of the institute to rest, then they received 5 days of handling. On the sixth day (baseline day), mice were placed into the first environmental context (environment "A") in a plexiglass foot-shocking chamber (25 cm × 25 cm × 30 cm) that was enriched with a specific combination of olfactory (macadamia nut scent), visual (striped walls), and tactile (metal bars on the floor) cues. Mice were allowed to move freely for 3 min to record baseline freezing levels (baseline light OFF period). Then, mice received 20-s long light stimulations 10 times (15 ms pulses at 25 Hz with 10 mW intensity, 473 nm, 10 s inter-light interval) to record baseline freezing levels during light stimulation period (baseline ON period). Mice displaying higher than 5% baseline freezing levels in environment "A" either during baseline light OFF or baseline light ON periods were excluded from the further analysis. On the seventh day (association day), mice were placed back into environment "A" and received light stimulations for 5 min (15 ms pulses at 25 Hz with 10 mW intensity, 473 nm, 10 s inter-light interval). A minute after the placement, mice received 4 foot shocks (2 s, 2 mA intensity, 58 s inter-shock interval) aligned with light illuminations. All mice displayed equally strong immediate reactions to foot shocks. After receiving the last shock, mice were kept in the context for another 1 min. After 4 successfully delivered shocks, mice were placed back into their home cages for 24 h. On the eighth day (readout day), mice were placed into the second environmental context (environment "B," 40 cm × 40 cm × 60 cm chamber) with distinct olfactory ("baby soap" scent), visual (brown curved walls and darker room lighting), and tactile (gray plastic floor) cues. Mice were allowed to move freely in the environment "B" for 5 min, where the last 2 min were recorded as the first OFF period (readout OFF period). Then, mice received a 3-min long laser light illumination (15 ms pulses at 25 Hz with 10 mW intensity, 473 nm), recorded as the readout ON period. After the termination of the light illumination, mice were kept in environment "B" for 2 min to record their post-light freezing levels (readout OFF period). We have shown that stimulation of NI hippocampal terminals could recall fear memories. This is highly likely to be a direct effect on local DG SOM neurons because NI targets them selectively and DG SOM neurons can produce the same effect. Furthermore, any antidromic effect (if it is possible at all in these long-range inhibitory cells) would be behaviorally negligible at such a long distance from the soma. However, even if it had such an unlikely effect, it would still confirm that NI HIPP-projecting GABAergic cells can recall fear memories at least partly with their hippocampal fibers.

## Optogenetic re-stimulation of NI GABAergic cells

After optic fiber implantations, mice were transferred to the animal room of the behavioral unit of the institute to rest, then they received 5 days of handling. On the sixth day, mice were placed into the first environmental context (environment "A," 20 cm × 20 cm × 20 cm chamber, with gray walls and striped floor, washed with macadamia soap) and were allowed to move freely for 3 min to record baseline freezing levels (baseline light OFF period). Then, mice

received light stimulations for 3 min (15 ms pulses at 25 Hz with 15 mW intensity, 473 nm) to record baseline freezing levels during light stimulation period (baseline light ON period). Mice displaying higher than 5% baseline freezing levels in environment "A" either during baseline light OFF or baseline light ON periods were excluded from further analysis. After recording the baseline periods mice were immediately placed into the second environmental context (environment "B") that was a plexiglass foot-shocking chamber (25 cm × 25 cm × 30 cm) enriched with a specific combination of olfactory ("baby soap" scent), visual (dim red room lighting), and tactile (metal bars on the floor) cues. Mice were allowed to move freely in environment "B" for 1 min. After this, mice received 4 foot shocks (2 s, 2 mA intensity, 58 s intershock interval) that were associated in time with 6-s long blue laser light illumination (15 ms pulses at 25 Hz with 15 mW intensity, 473 nm). Light illuminations were precisely aligned with the shocks, starting 2 s before the shock onset and finishing 2 s after shock offset. All mice displayed equally strong immediate reactions to foot shocks. After receiving the last shock, mice were kept in the context for another 1 min. After 4 successfully delivered shocks, mice were placed back into their home cages for 24 h. On the seventh day, mice were placed into the third environmental context (environment "C") in an open field chamber (40 cm × 36 cm × 15cm, washed with argan oil soap). Mice were allowed to move freely in environment "C" for 5 min, while the last 2 min was recorded as the first light OFF period (readout OFF period). Mice displaying higher than 5% freezing levels in this first light OFF period were excluded from the further analysis. Then, mice received a 3-min long blue laser light stimulation (15 ms pulses at 25 Hz with 15 mW intensity, 473 nm), recorded as readout light ON period. After the termination of the light illumination, mice were kept in environment "C" for 2 min to record post-light freezing levels (readout light OFF period).

## Behavioral experiment for c-Fos labeling in the NI and DG

For this experiment, we used vGAT-iRES-Cre/Gt(ROSA)26Sor-CAG/LSL-ZsGreen1 mice to identify all GABAergic cells without immunolabeling. A week before the experiments, mice were transferred to the animal room of the behavioral unit of the institute and then they received 5 days of handling. On the sixth day (preexposure day), mice were placed into environment "A" that was a plexiglass foot shocking chamber (25 cm × 25 cm × 30 cm) enriched with a specific combination of olfactory ("baby soap" scent), visual (striped walls), and tactile (metal bars on the floor) cues. Mice were allowed to freely move for 5 min, then were placed back into their home cages for 24 h. Mice displaying higher than 5% baseline freezing levels on the sixth day were excluded from the further analysis. On the seventh day, mice were placed back into environment "A," where they were allowed to freely move for 1 min, then they received 4 foot shocks (2 s, 2 mA intensity, 58 s inter-shock interval). All mice displayed equally strong immediate reactions to foot shocks. After receiving the last shock, mice were kept in the context for another 1 min. After 4 successfully delivered shocks, mice were placed back into their home cages for 24 h. On day 8, some mice were sacrificed from their home cages (Home-mice) to establish baseline c-Fos activity in the NI and in the DG. Whereas, some mice (Recent-mice) were placed back into environment "A" for 5 min that allowed them to recognize the context and we recorded their recent contextual fear memories. Then, Recent-mice were placed back into their home cages and 1 h after the context exposure, they were sacrificed to label c-Fos, and 30 days after the foot shocks (on the 37th day), some mice (Remote-mice) were placed back into environment "A" for 5 min to record their remote contextual fear memories. Then, Remote-mice were placed back into their home cages, and after 1 h, they were sacrificed to label c-Fos.

## Optogenetic inhibition of NI GABAergic cells during contextual fear recall

After optic fiber implantations, mice were transferred to the animal room of the behavioral unit of the institute to rest, then they received 5 days of handling. On the sixth day (baseline day), mice were placed into the first environmental context (environment "A") that was a plexiglass foot-shocking chamber (25 cm × 25 cm × 30 cm) enriched with a specific combination of olfactory (macadamia nut scent), visual (striped walls), and tactile (metal bars on the floor) cues. Mice were allowed to move freely for 1 min to record baseline freezing levels and then they received 4 foot shocks (2 s, 2 mA intensity, 58 s inter-shock interval). After receiving the last shock, mice were kept in the context for another 1 min. All mice displayed equally strong immediate reactions to foot shocks. After 4 successfully delivered shocks, mice were placed back into their home cages for 24 h. On the seventh day, mice were placed into the second environmental context (d7-envB, environment "B," 20 cm × 20 cm × 20 cm chamber) with mostly different environmental cues ("baby soap" scent, gray walls, dotted floor). After this, mice were replaced into the first environmental context (environment "A"), where we tested their contextual fear behavior on the seventh day (CFC7) and then mice were placed back into their home cages for 24 h. On the eighth day, mice were placed into environment "B," where we recorded their behavior again (d8-envB). After this, mice were placed back to the environment "A," where they immediately received a yellow laser light illumination (10 to 15 mW intensity at the tip of the optic fiber at 593 nm) and we recorded their contextual fear behavior on the eighth day (CFC8). We recorded and analyzed the first 3 min of each 4-min long readout period (d7-envB, d8-envB, CFC7, CFC8).

## Analysis and statistics for behavioral experiments

The behavior of mice was recorded with Basler acA1300-60gc camcorder and Noldus Ethovision 15.0 software and freezing behavior was analyzed manually using the Solomon Coder software (https://solomoncoder.com). The experimenters evaluating freezing levels were blind to the conditions and treatment of the mice. Motionless periods of mouse behavior (breathing only) were considered "freezing," but only if it lasted at least 2 s or more.

In case of data groups that did not show a Gaussian distribution, we used median and 25% to 75% interquartile range to present data. To test for statistical differences, we used the nonparametric Mann–Whitney U-test in independent data populations, and we used the Wilcoxon's signed-rank test in nonparametric dependent data populations. For the correlation analyses, we used nonparametric Spearman-rank correlation analyses. Statistical differences have always been tested using two-sided tests. Homogeneity of variance was tested using F-test and if it was significant then populations were compared using nonparametric tests. For indicating significance levels on figures, we used the following standard rules, *: $p < 0.05$, **: $p < 0.01$, ***: $p < 0.001$.

## Supporting information

**S1 Fig. Supplementary data for the re-inhibition of DG SOM cells.** (A) Three graphs show freezing behavior (% of total time) in each minute of the readout sessions (on day 8, in environment "C") for each mouse in experiments demonstrated in Fig 1, A to D: upper panel CTRL-mice, middle panel Associated eOPN3-mice, lower panel Not-associated eOPN3-mice. Data for CTRL-mice ($n = 10$, median [25%–75% quartiles]): first min: 12.00 [0.00–22.00], second min: 4.67 [0.00–8.00], third min: 0.00 [0.00–0.00], fourth min: 0.00 [0.00–4.00], fifth min: 0.00 [0.00–0.00], sixth min: 0.00 [0.00–4.00], seventh min: 0.00 [0.00–3.33]. Data for Associated eOPN3-mice ($n = 13$, median [25%–75% quartiles]): first min: 4.67 [0.00–15.00], second min: 4.67 [0.00–19.33], third min: 15.33 [3.67–34.00], fourth min: 12.67 [3.67–40.00], fifth

min: 16.33 [6.67–31.00], sixth min: 7.67 [0.00–10.67], seventh min: 0.00 [0.00–7.33]. Data for Not associated eOPN3-mice ($n$ = 8, median [25%–75% quartiles]): first min: 18.00 [3.33–25.33], second min: 4.00 [0.00–13.50], third min: 2.00 [0.00–6.00], fourth min: 0.00 [0.00–4.00], fifth min: 0.00 [0.00–3.67], sixth min.: 0.00 [0.00–2.33], seventh min: 0.00 [0.00–3.33]. Between-group statistics are labeled on graphs for Associated eOPN3-mice (red): comparison of the third minute period between CTRL and Associated eOPN3-mice: gray ***: $p$ = 0.0003, between Associated eOPN3 and Not-associated eOPN3-mice: pink *: $p$ = 0.027; comparison of the fourth minute period between CTRL and Associated eOPN3-mice: gray *: $p$ = 0.012, between Associated eOPN3 and Not-associated eOPN3-mice: pink *: $p$ = 0.019; comparison of the fifth minute period between CTRL and Associated eOPN3-mice: gray **: $p$ = 0.006, Associated eOPN3 and Not-associated eOPN3-mice: pink **: $p$ = 0.004 (Mann–Whitney U-tests). (B) Three graphs show freezing behavior (% of total time) in 30 s right before and right after the start of the light illumination. Data for CTRL-mice ($n$ = 10, median [25%–75% quartiles]): last 30 s before: 0.00 [0.00–7.33], first 30 s after: 0.00 [0.00–0.00]. Statistics: comparison of 30 s before to 30 s after: n.s.: non-significant, $p$ = 0.109, (Wilcoxon signed-rank test). Data for Associated eOPN3-mice ($n$ = 13, median [25%–75% quartiles]): last 30 s before: 7.33 [0.00–10.00], first 30 s after: 23.33 [6.67–40.67]. Statistics: comparison of 30 s before to 30 s after: black *: $p$ = 0.021, (Wilcoxon signed-rank test). Data for Not-associated eOPN3-mice ($n$ = 8, median [25%–75% quartiles]): last 30 s before: 3.67 [0.00–11.67], first 30 s after: 0.00 [0.00–0.00]. Statistics: comparison of 30 s before to 30 s after: n.s.: non-significant, $p$ = 0.500, (Wilcoxon signed-rank test). Between-group statistics are labeled on graphs for Associated eOPN3-mice (red): comparison of the first 30 s period after the start of the illumination between CTRL and Associated eOPN3-mice: gray ***: $p$ = 0.0007; comparison of the same between Not-associated eOPN3 and Associated eOPN3-mice: pink *: $p$ = 0.014 (Mann–Whitney U-tests). (C) Graph shows freezing time during baseline 3–3 min light OFF and light ON periods on day 6, and freezing time during 2-3-2 min light OFF, ON and OFF periods on day 8, respectively. Data for CTRL-mice ($n$ = 10, median [25%–75% quartiles]): baseline OFF: 0.00 [0.00–0.00], baseline ON: 0.00 [0.00–0.00], readout OFF: 8.17 [3.50–15.50], readout ON: 1.50 [0.00–3.33], readout OFF: 0.83 [0.00–2.00]. Data for Associated eOPN3-mice ($n$ = 13, median [25%–75% quartiles]): baseline OFF: 0.00 [0.00–0.00], baseline ON: 0.00 [0.00–0.00], readout OFF: 7.50 [0.00–16.33], readout ON: 15.00 [5.11–33.67], readout OFF: 3.83 [1.67–7.83]. Data for Not-associated eOPN3-mice ($n$ = 8, median [25%–75% quartiles]): baseline OFF: 0.00 [0.00–0.00], baseline ON: 0.00 [0.00–0.01], readout OFF: 12.00 [7.17–14.83], readout ON: 2.11 [0.00–6.22], readout OFF: 0.83 [0.00–2.75]. Between-group statistics: comparison of the readout first OFF periods between CTRL and Associated eOPN3-mice: n.s.: non-significant, $p$ = 0.778; between Associated eOPN3-mice and Not-associated eOPN3-mice: n.s.: non-significant, $p$ = 0.466 (Mann–Whitney U-tests). Comparison of the readout ON periods between CTRL and Associated eOPN3-mice: **: $p$ = 0.001, between Associated eOPN3-mice and Not-associated eOPN3-mice: **: $p$ = 0.007 (Mann–Whitney U-tests). Comparison of the readout second OFF periods between CTRL and Associated eOPN3-mice: n.s.: non-significant, $p$ = 0.128, between Associated eOPN3-mice and Not-associated eOPN3-mice: n.s.: non-significant, $p$ = 0.075 (Mann–Whitney U-tests). (D) Graph shows individual freezing time during 3-3-3 min light OFF, ON, OFF periods on day 8, and freezing time during 3 min contextual fear readout period in environment "B" on day 9 for Associated eOPN3-mice ($n$ = 5) and Not associated eOPN3-mice ($n$ = 8). Brown lines represent the medians for each periods during the 2 days. Data for Associated eOPN3-mice ($n$ = 5, median [25%–75% quartiles]): readout OFF: 12.50 [7.50–19.17], readout ON: 38.11 [13.22–47.44], readout OFF: 3.83 [3.67–7.83], contextual fear readout: 43.33 [34.00–52.00]. Statistics: comparison of the first readout OFF period to contextual fear readout period: *: $p$ = 0.043, (Wilcoxon signed-rank test). Data for Not associated

eOPN3-mice ($n$ = 8, median [25%–75% quartiles]): readout OFF: 12.00 [7.17–14.83], readout ON: 2.11 [0.00–6.22], readout OFF: 0.83 [0.00–2.75], contextual fear readout: 50.39 [29.00–69.17]. Statistics: comparison of the first readout OFF period to contextual fear readout period: *: $p$ = 0.012, (Wilcoxon signed-rank test). Between-group statistics: comparison of the contextual fear readout periods between Associated eOPN3-mice and Not-associated eOPN3-mice: n.s.: non-significant, $p$ = 0.608 (Mann–Whitney U-tests). (E) Graphs show freezing behavior (% of total time) in each minute of the contextual fear memory readout on day 7 for CTRL ($n$ = 12) and hM4Di ($n$ = 11) mice, in experiments demonstrated in Fig 1H and 1I: Data for CTRL-mice ($n$ = 12, median [25%–75% quartiles]): first min: 16.33 [7.50–28.50], second min: 35.00 [20.83–46.33], third min: 31.50 [12.50–38.50]. Data for hM4Di-mice ($n$ = 11, median [25%–75% quartiles]): first min: 0.00 [0.00–4.00], second min: 7.00 [0.00–15.67], third min: 12.33 [4.33–18.00]. Statistics: comparison of CTRL vs. hM4Di-mice: first min.: ***: $p$ = 0.0007, second min: **: $p$ = 0.003, third min: *: $p$ = 0.029 (Mann–Whitney U-tests). (F) Fluorescent images show the specificity of the used AAV2/1-hSyn-SIO-eOPN3-mScarlet virus. The upper image from a wild-type (WT) mouse ($n$ = 2 mice) show no viral expression anywhere in hippocampus or the DG (shown). The middle image from SOM-Cre mouse show viral expression typical of SOM cells in the DG, and the lower image from PV-Cre mouse show viral expression typical of PV cells in the DG. Scale bar: 200 μm. (G) Representative fluorescent images from DG show that eOPN3-mScarlet-labeled SOM cells (red) are indeed immunopositive for SOM immunostaining. Double immunopositive cells are labeled with white arrowheads. At least 92% (189/205) of DG eOPN3-mScarlet-labeled SOM cells were clearly immunopositive for SOM and rest of them were only faintly positive ($n$ = 4 eOPN3-mice). Individually these data are the following: mouse1 (43/46), mouse2 (53/56), mouse3 (22/25), mouse4 (71/78). At least 55% (184/335) of DG immunolabeled SOM cells were infected with eOPN3-mScarlet-containing AAV in the middle of the injection site below the optic fibers. Scale bar: 20 μm. Individually these data are the following: mouse1 (44/98), mouse2 (46/92), mouse3 (22/36), mouse4 (72/109). The data underlying this figure can be found in S1 Data.
(TIF)

**S2 Fig. Inhibition of DG SOM cells needs to be temporally precise to recall fear memories efficiently and we show supplementary data for the re-inhibition of DG PV cells and CA1 SOM cells.** (A) After infecting DG SOM cells with inhibitory ArchT opsin-containing AAVs bilaterally, we implanted optic fibers over DGs. Representative image: injection site and optic fiber position (yellow). Scale bar: 200 μm. Day 6, in environment "A," "Align. mice" received foot shocks aligned with light illumination, while "Shift. mice" received light illumination 60 s (±2 s) after each foot shocks. (B) Day 7, in environment "B": 2 min OFF—3 min ON—2 min OFF light cycle. "Align. mice" (dark green) could recall fear memory significantly more efficiently. These ArchT mediated recalls in environment B are significant but less effective than those mediated by eOPN3 that had a longer deactivation time during fear conditioning (Fig 1). The graph shows freezing time during the 3 min light ON period on day 7 in environment "B" (medians and interquartile ranges). Data for "Shift. mice" ($n$ = 8, median [25%–75% quartiles]): readout ON: 2.62 [0.63–5.16]. Data for "Align. mice" ($n$ = 7, median [25%–75% quartiles]): readout ON: 6.90 [4.18–11.39]. Between-group statistics: comparison of the readout first OFF periods between Shift. and Align. mice: n.s.: non-significant, $p$ = 0.954; comparison of the readout ON periods between Shift. and Align. mice: *: $p$ = 0.024; comparison of the readout second OFF periods between Shift. and Align. mice: n.s.: non-significant, $p$ = 0.321 (Mann–Whitney U-tests). (C) Day 8, in environment "A": 2 min OFF - 3min ON—2 min OFF light cycle. Light illumination is significantly more efficient in "Align. mice." Graph shows freezing time during 3 min light ON period on day 8 in environment "A" (medians and

interquartile ranges). Data for "Shift. mice" ($n = 8$, median [25%–75% quartiles]): readout ON: 11.02 [6.26–14.52]. Data for "Align. mice" ($n = 7$, median [25%–75% quartiles]): readout ON: 46.77 [27.80–60.19]. Between-group statistics: comparison of the readout first OFF periods between Shift. and Align. mice: n.s.: non-significant, $p = 0.148$; comparison of the readout ON periods between Shift. and Align. mice: **: $p = 0.003$; comparison of the readout second OFF periods between Shift. and Align. mice: *: $p = 0.043$ (Mann–Whitney U-tests). (D) Graph shows freezing time during baseline 3–3 min light OFF and ON periods on day 6, and freezing time during 2-3-2 min light OFF, ON and OFF periods on day 8 for PV-Cre CTRL ($n = 8$) and eOPN3 ($n = 9$) mice. Data for CTRL-mice ($n = 8$, median [25%–75% quartiles]): baseline OFF: 0.00 [0.00–0.61], baseline ON: 0.00 [0.00–0.56], readout OFF: 5.42 [0.00–7.58], readout ON: 0.00 [0.00–2.11], readout OFF: 0.92 [0.00–2.67]. Data for eOPN3-mice ($n = 9$, median [25%–75% quartiles]): baseline OFF: 0.00 [0.00–0.00], baseline ON: 0.00 [0.00–0.00], readout OFF: 0.00 [0.00–1.83], readout ON: 0.00 [0.00–0.00], readout OFF: 0.00 [0.00–0.00]. Between-group statistics: comparison of the first readout OFF periods between CTRL and eOPN3-mice: n.s.: non-significant, $p = 0.322$; comparison of the readout ON periods between CTRL and eOPN3-mice: n.s.: non-significant, $p = 0.219$; comparison of the second readout OFF periods between CTRL and eOPN3-mice: n.s.: non-significant, $p = 0.158$ (Mann–Whitney U-tests). (E) Graph shows freezing time during baseline 3–3 min light OFF and ON periods on day 6, and freezing time during 2-3-2 min light OFF, ON and OFF periods on day 8 for CA1-injected SOM-Cre CTRL ($n = 6$) and ArchT ($n = 11$) mice. Data for CTRL-mice ($n = 6$, median [25%–75% quartiles]): baseline OFF: 0.00 [0.00–0.61], baseline ON: 0.00 [0.00–0.00], readout OFF: 22.67 [3.50–31.00], readout ON: 4.06 [0.00–6.67], readout OFF: 4.75 [1.67–15.50]. Data for ArchT-mice ($n = 11$, median [25%–75% quartiles]): baseline OFF: 0.00 [0.00–0.61], baseline ON: 0.00 [0.00–0.00], readout OFF: 7.33 [4.33–34.67], readout ON: 16.89 [6.89–30.44], readout OFF: 6.17 [1.83–16.33]. Between-group statistics: comparison of the first readout OFF periods between CTRL and ArchT-mice: n.s.: non-significant, $p = 0.725$; comparison of the readout ON periods between CTRL and ArchT-mice: *: $p = 0.018$; comparison of the second readout OFF periods between CTRL and ArchT-mice: n.s.: non-significant, $p = 0.801$ (Mann–Whitney U-tests). (F) Graphs show freezing levels in each minute (% of total time) during the readout session (panel B) for CTRL and ArchT-mice. Data for CTRL-mice ($n = 6$, median [25%–75% quartiles]): first min: 33.00 [3.33–53.00], second min: 6.33 [3.33–21.33], third min: 0.00 [0.00–8.00], fourth min: 0.00 [0.00–14.33], fifth min: 0.00 [0.00–4.00], sixth min: 4.17 [3.33–13.00], seventh min: 5.33 [0.00–11.33]. Data for ArchT-mice ($n = 11$, median [25%–75% quartiles]): first min: 11.33 [4.00–46.00], second min: 4.67 [4.00–21.00], third min: 18.00 [3.33–27.33], fourth min: 11.33 [4.33–26.67], fifth min: 17.00 [8.67–39.33], sixth min: 4.00 [0.00–10.67], seventh min: 8.00 [0.00–23.67]. Between-group statistics: comparison of the third minute period between CTRL and ArchT-mice: $p = 0.064$ (purple); comparison of the fourth minute period between CTRL and ArchT-mice: purple *: $p = 0.048$; comparison of the fifth minute period between CTRL and ArchT-mice: purple *: $p = 0.011$ (Mann–Whitney U-tests). The data underlying this figure can be found in S1 Data.
(TIF)

**S3 Fig. Monosynaptic inputs of DG SOM cells.** (A) Cre-dependent helper viruses were injected into the dorsal and ventral DG bilaterally of SOM-Cre mice, followed by an injection of Rabies(ΔG)-EnvA-eGFP 5 weeks later ($n = 2$ mice). (B) Injection site of helper (red) and rabies (green) viruses in the DG. White arrowheads show the starter cells (that express both viruses) in the hilus. Scale bar: 200 μm. (C) Rabies-labeled neurons in different intrahippocampal as well as subcortical brain areas establish synapses on DG SOM-positive neurons. Scale bar: 200 μm. HDB: horizontal diagonal band of Broca, MS/VDB: medial septum/vertical

diagonal band of Broca, NI: nucleus incertus, MRR: median raphe region. (D–G) Fluorescent images show that DG SOM cells innervating rabies infected input neurons were clearly positive for PV (6/45) in MS/VDB (D), or positive for ChAT (30/45) in MS/VDB (E), or positive for Relaxin-3 (3/3) in NI (F), or positive for vGluT3 (7/9) in MRR (G). Scale bar: 20 μm for every images.
(TIF)

**S4 Fig. Supplementary data for the in vivo electrophysiology experiments.** (A) Confocal laser scanning microscopic images from virally injected SOM-Cre/vGAT-Flp mouse (described in Fig 3A) show the location of NI GABAergic fibers (green) in the DG. Arrowheads show putative synaptic contacts of NI fibers onto DG SOM cells (red). Scale bar: 200 μm. (B) Local field potential recordings of a dentate spike at different depth in the hippocampal formation. Right: amplitude of the dentate spike at different depths. Note the reversal of the amplitude. (C) Single units are plotted based on their trough to peak interval and burst indices (the latter is on a logarithmic scale). Dashed horizontal line indicate the separation between the putative excitatory (red) and putative inhibitory units (blue). Putative inhibitory neurons were defined if the burst index was less than 4, while putative excitatory neurons were defined if the burst index was more than 4 (*72*). Right insets depict sample average waveforms of a putative excitatory (top) and inhibitory units (bottom) from the DG with the corresponding auto-correlograms below them. The data underlying this figure can be found in S1 Data.
(TIF)

**S5 Fig. Supplementary data for the reactivation of NI GABAergic fibers in DG or somata of the cells in the NI.** (A) Graph shows freezing time during the 3 min baseline light OFF and 5 min light ON periods on day 6, and freezing time during 2-3-2 min light OFF, ON and OFF periods on day 8, respectively. Data for CTRL-mice ($n$ = 12, median [25%–75% quartiles]): baseline OFF: 0.00 [0.00–0.00], baseline ON: 0.00 [0.00–0.00], readout OFF: 5.08 [2.00–14.58], readout ON: 3.56 [0.00–7.94], readout OFF: 1.00 [0.00–8.67]. Data for ChR2 mice ($n$ = 12, median [25%–75% quartiles]): baseline OFF: 0.00 [0.00–0.00], baseline ON: 0.00 [0.00–0.00], readout OFF: 8.92 [4.17–16.92], readout ON: 14.72 [9.78–22.17], readout OFF: 7.08 [2.67–12.75]. Between-group statistics: comparison of the readout first OFF periods between CTRL and ChR2 mice: n.s.: non-significant, $p$ = 0.488; comparison of the readout ON periods between CTRL and ChR2 mice: ***: $p$ = 0.0007; comparison of the readout second OFF periods between CTRL and ChR2 mice: n.s.: non-significant, $p$ = 0.112 (Mann–Whitney U-tests). (B) Graphs show freezing levels (% of total time) in each minute during the readout session (also in panel A) for CTRL ($n$ = 12) and ChR2 ($n$ = 12) mice. Data for CTRL-mice ($n$ = 12, median [25%–75% quartiles]): first min: 6.33 [4.00–16.00], second min: 2.33 [0.00–9.83], third min: 5.00 [0.00–9.33], fourth min: 0.00 [0.00–10.33], fifth min: 0.00 [0.00–7.50], sixth min: 1.67 [0.00–5.33], seventh min: 0.00 [0.00–5.83]. Statistics: comparison of the third min and fourth min: n.s.: non-significant, $p$ = 0.575 (Wilcoxon signed-rank test). Data for ChR2 mice ($n$ = 12, median [25%–75% quartiles]): first min: 6.17 [3.33–14.67], second min: 9.50 [4.83–15.33], third min: 14.67 [8.00–22.67], fourth min: 16.33 [10.33–20.83], fifth min: 16.17 [4.50–22.17], sixth min: 7.17 [2.50–13.17], seventh min: 6.33 [0.00–12.50]. Statistics: comparison of the second min and third min: black *: $p$ = 0.041 (Wilcoxon signed-rank test). Between-group statistics: comparison of the third minute period between CTRL and ChR2 mice: purple **: $p$ = 0.009; comparison of the fourth minute period between CTRL and ChR2 mice: purple **: $p$ = 0.004; comparison of the fifth minute period between CTRL and ChR2 mice: purple **: $p$ = 0.002 (Mann–Whitney U-tests). (C) Graph shows freezing time during 3–3 min baseline light OFF and ON periods on day 6, and freezing time during 2-3-2 min light OFF, ON and OFF periods on day 7. Data for CTRL-mice ($n$ = 11, median [25%–75% quartiles]): baseline

OFF: 0.00 [0.00–0.00], baseline ON: 0.00 [0.00–1.28], readout OFF: 0.00 [0.00–1.83], readout ON: 0.00 [0.00–4.11], readout OFF: 1.83 [0.00–2.33]. Data for ChR2 mice ($n = 7$, median [25%–75% quartiles]): baseline OFF: 0.00 [0.00–0.00], baseline ON: 0.00 [0.00–0.00], readout OFF: 1.83 [0.00–3.67], readout ON: 16.44 [8.00–18.56], readout OFF: 5.17 [0.00–12.50]. Between-group statistics: comparison of the first readout OFF periods between CTRL and ChR2 mice: n.s.: non-significant, $p = 0.254$; comparison of the readout ON periods between CTRL and ChR2 mice: **: $p = 0.0012$; comparison of the second readout OFF periods between CTRL and ChR2 mice: n.s.: non-significant, $p = 0.161$ (Mann–Whitney U-tests). (D) Graphs show freezing levels (% of total time) in each minute during the readout session (also in panel C) for CTRL ($n = 11$) and ChR2 ($n = 7$) mice. Data for CTRL-mice ($n = 11$, median [25%–75% quartiles]): first min: 0.00 [0.00–3.67], second min: 0.00 [0.00–0.00], third min: 0.00 [0.00–3.67], fourth min: 0.00 [0.00–3.67], fifth min: 0.00 [0.00–3.33], sixth min: 3.67 [0.00–4.00], seventh min: 0.00 [0.00–3.33]. Statistics: comparison of the second min and third min: n.s.: non-significant, $p = 0.144$, (Wilcoxon signed-rank test). Data for ChR2 mice ($n = 7$, median [25%–75% quartiles]): first min: 3.33 [0.00–7.33], second min: 0.00 [0.00–3.33], third min: 15.00 [3.33–24.67], fourth min: 16.33 [9.00–25.67], fifth min: 10.00 [3.33–20.33], sixth min.: 0.00 [0.00–11.00], seventh min: 3.67 [0.00–14.00]. Statistics: comparison of the second min and third min: black *: $p = 0.018$ (Wilcoxon signed-rank test). Between-group statistics: comparison of the third minute period between CTRL and ChR2 mice: purple **: $p = 0.004$; comparison of the fourth minute period between CTRL and ChR2 mice: purple **: $p = 0.003$; comparison of the fifth minute period between CTRL and ChR2 mice: purple **: $p = 0.009$ (Mann–Whitney U-tests). The data underlying this figure can be found in S1 Data. (TIF)

**S6 Fig. Supplementary data for the natural reactivation NI GABAergic cells during contextual fear memory recall and a behavioral control experiment.** (A, B) Scatterplot shows a significant correlation (Spearman-rank correlation) between the density of c-Fos labeled vGAT positive cells (cells/mm2) in NI and the density of c-Fos labeled cells (cells/mm2) in DG granule cell layer in 10 Recent mice. Representative fluorescent images show c-Fos positive cells (red) in the DG and in the NI from 2 mice the data point of which are labeled with a green circle around the original data point in panel A. These fluorescent images illustrate that the more c-Fos positive GABAergic (green) cells were observed in the NI (lower panel), the more c-Fos positive cells were observed in the granule cell layer of the DG (upper panel) in 10 Recent mice that successfully recalled their memory. Scale bars: 200 μm for DG images, 50 μm for NI images. s.gr.: stratum granulosum, hil: hilus. (C) Scatterplot shows no correlation between the density of c-Fos labeled vGAT positive cells (cells/mm2) in NI and the density of c-Fos labeled cells (cells/mm2) in DG granule cell layer in 8 control "Homecage mice." Correlation details are as follows: R = −0.214, non-significant: $p = 0.610$ (Spearman-rank correlation). (D) Differences in density (cells/mm2) of c-Fos positive cells in DG GC layer in Homecage mice and Recent mice (medians and interquartile ranges). Data for Homecage mice ($n = 8$, median [25%–75% quartiles]): 268.44 [245.24–327.79]. Data for Recent mice ($n = 10$, median [25%–75% quartiles]): 477.13 [401.98–524.43]. Statistics: **: $p = 0.005$ (Mann–Whitney U-test). (E) The whole behavioral paradigm of the inhibition of NI cells during contextual memory recall. This illustration explains how fear behavior was tested in a control environment "B" (Fig 5). (F) Graph shows individual data points of the time spent with freezing behavior in environment "B" for each mouse (median [25%–75% quartiles]) on day 7 (d7-envB) and day 8 (d8-envB), respectively (also see panel D). Data for CTRL-mice in d7-envB: 11.56 [8.83–17.33], in d8-envB: 14.72 [1.94–16.28] and $n = 12$. Data for ArchT-mice in d7-envB: 16.11 [10.83–27.44], in d8-envB: 14.61 [7.22–21.56] and $n = 12$. Statistics: CTRL-mice in d7-envB vs.

d8-envB: n.s.: non-significant: $p$ = 0.594, (Wilcoxon signed-rank test). CTRL-mice d7-envB vs. ArchT-mice d7-envB: n.s.: non-significant: $p$ = 0.341, (Mann–Whitney U-test). ArchT-mice in d7-envB vs. d8-envB: n.s.: non-significant: $p$ = 0.099 (Wilcoxon signed-rank test). The data underlying this figure can be found in S1 Data.
(TIF)

**S7 Fig. Supplementary data for cortical innervation of NI.** (A) Representative fluorescent images show that NI is innervated by cortical areas differently and illustrates how NI areas were equally divided into 10 sub-areas for measurement. Scale bar: 200 μm. (B) Three graphs show relative pixel intensities for each sub-areas of NI innervated by the PFC, ACC, and RSC, respectively (medians and interquartile ranges). Statistics: 2 mice per group with 12 (PFC) or 10 (ACC, RSC) individual data points per sub-area.*: $p < 0.05$, **$p < 0.01$, ***: $p < 0.001$ (multiple comparisons with Bonferroni-corrected $p$-values). Blue stars indicate significance between PFC and RSC groups or ACC and RSC groups, red stars indicate significance between ACC and PFC groups. The data underlying this figure can be found in S1 Data.
(TIF)

**S8 Fig. Injection sites and optic fiber localizations in the HIPP.** Summary of virus injection sites in the HIPP in every mouse used in the behavioral opto- and chemogenetic experiments. The virus injection sites in different mice from all experiments were analyzed. We localized the exact regions were traces of the virus injection in each section of the region could be detected. Then, we illustrated these areas in the stereological atlas (Paxinos G, Franklin K. 2008. Mouse brain in stereotaxic coordinates). Then, we overlaid these illustrations on each other to get a representative localization of the region of the viral infection around the tip of the optic fiber or around the middle of the injection sites. (A1-3) Images show AAV2/2-EF1α-DIO-mCherry (A1, brown) and AAV2/1-hSyn-SIO-eOPN3-mScarlet (A2-3, pink) virus injection sites and positions of the optic fibers (gray, red, and pink fibers) for mice used in DG SOM cells re-inhibition experiment (described in Fig 1A). Images represent the 10 injection sites of CTRL-mice (A1), 13 injection sites of Associated eOPN3-mice (A2, blue dots in the hilus represent somata of the illuminated cells), and 8 injection sites of Not associated eOPN3-mice (A3). (B1-2) Images show AAV2/2-EF1α-DIO-mCherry (B1, brown) and AAV2/8-hSyn-DIO-hM4D(Gi)-mCherry (B2, pink) virus injection sites for mice used in chemogenetic experiment with DG SOM cells (described in Fig 1H). Images represent the 12 injection sites of CTRL-mice (B1) and 11 injection sites of hM4Di-mice (B2). (C1) Images show AAV2/1-hSyn-SIO-eOPN3-mScarlet (pink) virus injection sites and positions of the optic fibers for eOPN3-mice ($n$ = 6) used in optogenetic c-Fos staining experiment with DG SOM cells (described in Fig 1E). Blue fibers represent the illuminated sides; gray fibers represent the non-illuminated sides. (D1-2) Images show AAV2/2-EF1α-DIO-mCherry (D1, brown) and AAV2/1-hSyn-SIO-eOPN3-mScarlet (D2, pink) virus injection sites and positions of the optic fibers (gray and red fibers) for mice used in DG PV cells re-inhibition experiment (described in Fig 2A). Images represent the 8 injection sites of CTRL-mice (D1) and 9 injection sites of eOPN3-mice (D2). E1-2: Images show AAV2/5-CAG-FLEX-ArchT-GFP in shifted group (E1, light green) and AAV2/5-CAG-FLEX-ArchT-GFP in aligned group (E2, green) virus injection sites and positions of the optic fibers (gray and orange fibers) for mice used in DG SOM cells shifted or aligned inhibition experiment (described in S2 Fig.). Images represent the 8 injection sites of shifted-mice (E1) and 7 injection sites of aligned-mice (E2). (F1-2) Images show AAV2/5-EF1α-DIO-eYFP (F1, light green) and AAV2/5-CAG-FLEX--ArchT-GFP (F2, green) virus injection sites and positions of the optic fibers (gray and orange fibers) for mice used in CA1 SOM cells re-inhibition experiment (described in Fig 2E). Images represent the 6 injection sites of CTRL-mice (F1), and 11 injection sites of ArchT-mice (F2).

(G1-2) Images show the position of the optic fibers for mice used in optogenetic c-Fos experiment of NI DG-projecting fibers (described in Fig 3C). Blue fibers represent the illuminated sides; grey fibers represent the non-illuminated sides. Images represent the CTRL-mice (G1, *n* = 8) and ChR2 mice (G2, *n* = 5). (H1-2) Images show the position of the optic fibers (gray and blue fibers) for mice used in NI DG-projecting fibers restimulation experiment (described in Fig 3F). Images represent CTRL-mice (H1, *n* = 12) and ChR2 mice (H2, *n* = 12).
(TIF)

**S9 Fig. Injection sites and optic fibers localizations in the NI.** Summary of virus injection sites in the NI in every mouse used in the behavioral optogenetic experiments. The virus injection sites in different mice from all experiments were analyzed. We localized the exact regions where traces of the virus injection in each section of the region could be detected. Then, we illustrated these areas in the stereological atlas (Paxinos G, Franklin K. 2008. Mouse brain in stereotaxic coordinates). Then, we overlaid these illustrations on each other to get a representative localization of the region of the viral infection around the tip of the optic fibers or around the middle of the injection sites. The positions of the tips are represented by the lowest points of the geometric forms. (A1-2) Images show AAV2/5-EF1α-DIO-eYFP (A1, light green) and AAV2/5-EF1α-DIO-ChR2-eYFP (A2, green) virus injection sites for mice used in optogenetic c-Fos experiment of NI DG-projecting fibers (described in Fig 3C). Images represent the 8 injection sites of CTRL-mice (A1) and 5 injection sites of ChR2 mice (A2). (B1-2) Images show AAV2/5-EF1α-DIO-eYFP (B1, light green) and AAV2/5-EF1α-DIO-ChR2-eYFP (B2, green) virus injection sites for mice used in NI DG-projecting fibers re-stimulation experiment (described in Fig 3F). Images represent the 12 injection sites of CTRL-mice (B1) and 12 injection sites of ChR2 mice (B2). (C1-2) Images show AAV2/5-EF1α-DIO-eYFP (C1, light green) and AAV2/5-EF1α-DIO-ChR2-eYFP (C2, green) virus injection sites and positions of the tip of the optic fibers (blue circles, blue diamonds) for mice used in NI cells re-stimulation experiment (described in Fig 4A). Images represent the 11 injection sites of CTRL-mice (C1) and 7 injection sites of ChR2 mice (C2). (D1-2) Images show AAV2/5-EF1α-DIO-eYFP (D1, light green) and AAV2/5-CAG-FLEX-ArchT-GFP (D2, green) virus injection sites and positions of the tip of the optic fibers (orange circles, orange diamonds) for mice used in NI cell inhibition experiment (described in Fig 5G). Images represent the 12 injection sites of CTRL-mice (D1) and 12 injection sites of ArchT-mice (D2).
(TIF)

**S10 Fig. Schematic illustration of the model: engram cell formation and reactivation in the hippocampus (CA1 and DG).** The hippocampus receives contextual fear processing-related excitatory inputs from cortical areas partly via the entorhinal cortex (A, B, purple) that broadcasts a rough, only partly context-specific activation pattern to the dendrites of principal neurons (*15*). This gives some principal cells only the ***opportunity*** to become a memory trace-encoding engram cell, because most principal cells are kept inhibited by dendrite-targeting SOM cells, whereas others do not even get strong enough inputs. On the other hand, the hippocampus receives inhibitory inputs from the brainstem nucleus incertus (NI) as well that initiates fear memory-specific disinhibition of principal neurons via local hippocampal SOM cells (B, green). This gives an overlapping population of principal cells ***permission*** to become a memory trace-encoding engram cell (red). This mechanism may be initiated by memory-processing neocortical centers (mPFC, ACC, RSC) via both the entorhinal cortex and the NI simultaneously. The interaction of these (and other modulatory) inputs may provide the basic mechanism for both the formation and reactivation of engram cells in the hippocampus region. (Dark colors illustrate active and light colors illustrate inactive fibers, cells or dendritic

compartments.).
(TIF)

**S1 Table. Characterization of used primary antibodies and retrograde tracers.**
(XLSX)

**S2 Table. Secondary antibodies.**
(XLSX)

**S3 Table. Primary and secondary antibody combinations used in immunofluorescent experiments.**
(XLSX)

**S1 Data. All data points for all graphs in all figures.**
(XLSX)

**S1 Extended Data. Extended Data for Main Fig 1.**
(DOCX)

**S2 Extended Data. Extended Data for Main Fig 2.**
(DOCX)

**S3 Extended Data. Extended Data for Main Fig 3.**
(DOCX)

**S4 Extended Data. Extended Data for Main Fig 4.**
(DOCX)

**S5 Extended Data. Extended Data for Main Fig 5.**
(DOCX)

**S6 Extended Data. Extended Data for Main Fig 6.**
(DOCX)

## Acknowledgments

We would like to thank M. Mahn and O. Yizhar for their help with the AAV2/1-hSyn-SIO-eOPN3-mScarlet virus. We thank J. Huang for helping with the SOM-Cre mice. We thank A. L. Gundlach for providing the anti-Relaxin-3 antibody. We would like to thank M. Sümegi and E. Sipos, the Virus Technology Unit of IEM, for technical support. We would like to thank P. Vági and L. Barna, the Nikon Microscopy Center at IEM, Nikon Austria GmbH, and Auro-Science Consulting, for imaging technical support. We thank K. Demeter and the Behavior Studies Unit of the IEM for behavioral experiment support. We thank Z. Erdélyi, F. Erdélyi, and the staff of the Animal Facility and the Medical Gene Technology Unit of IEM for expert technical help with the breeding and genotyping of the mouse strains used in this study. We would like to thank Z. Hajós, E. Szépné Simon, and N. Kriczky for helping with the experiments; and A. Kriczky, K. Iványi, and G. Goda for other assistance. We thank L. Acsády, T.F. Freund, S. Káli, and A. Szőnyi for their comments on an earlier version of this manuscript.

## Author Contributions

**Conceptualization:** Krisztián Zichó, Albert M. Barth, Gábor Nyiri.

**Data curation:** Krisztián Zichó, Péter Papp, Márton I. Mayer, Gábor Nyiri.

**Formal analysis:** Krisztián Zichó, Péter Papp, Albert M. Barth, Márton I. Mayer, Réka Z. Sebestény, Gábor Nyiri.

**Funding acquisition:** Gábor Nyiri.

**Investigation:** Krisztián Zichó, Katalin E. Sos, Péter Papp, Albert M. Barth, Erik Misák, Áron Orosz, Márton I. Mayer, Réka Z. Sebestény.

**Project administration:** Krisztián Zichó, Péter Papp, Gábor Nyiri.

**Resources:** Gábor Nyiri.

**Supervision:** Gábor Nyiri.

**Validation:** Krisztián Zichó, Márton I. Mayer, Gábor Nyiri.

**Visualization:** Krisztián Zichó, Péter Papp, Gábor Nyiri.

**Writing – original draft:** Krisztián Zichó, Albert M. Barth, Gábor Nyiri.

**Writing – review & editing:** Krisztián Zichó, Katalin E. Sos, Péter Papp, Albert M. Barth, Erik Misák, Áron Orosz, Márton I. Mayer, Réka Z. Sebestény, Gábor Nyiri.

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
