## [Editor Report · Decision Letter 0]

23 Sep 2022

Dear Dr Nyiri, 

Thank you for submitting your manuscript entitled "Fear memory recall via hippocampal somatostatin interneurons" for consideration as a Research Article by PLOS Biology.

Your manuscript has now been evaluated by the PLOS Biology editorial staff, as well as by an academic editor with relevant expertise, and I am writing to let you know that we would like to send your submission out for external peer review.

Once your full submission is complete, your paper will undergo a series of checks in preparation for peer review. After your manuscript has passed the checks it will be sent out for review. To provide the metadata for your submission, please Login to Editorial Manager (https://www.editorialmanager.com/pbiology) within two working days, i.e. by Sep 25 2022 11:59PM.

I also apologize for the length of time it took me to get this decision out to you. I was traveling last week and our Academic Editor was busy earlier this week - so it took a bit of time before the Academic Editor and I could connect.

Kind regards,

Kris

Kris Dickson, Ph.D. (she/her)

Neurosciences Senior Editor/Section Manager

PLOS Biology

kdickson@plos.org

---

## [Decision Letter · Decision Letter 1]

16 Nov 2022

Dear Dr Nyiri,

Thank you for your patience while your manuscript "Fear memory recall via hippocampal somatostatin interneurons" was peer-reviewed at PLOS Biology. Your manuscript has been evaluated by the PLOS Biology editors, an Academic Editor with relevant expertise, and by several independent reviewers.

As you will see in the reviewer reports, which can be found at the end of this email, although the reviewers find the work potentially interesting, they have also raised a number of important concerns. Based on their specific comments and following discussion with the Academic Editor, it is clear that a substantial amount of work would be required to meet the criteria for publication in PLOS Biology. However, given our and the reviewer interest in your study, we would be open to inviting a comprehensive revision of the study that thoroughly addresses all the reviewers' comments. Given the extent of revision that would be needed, we cannot make a decision about publication until we have seen the revised manuscript and your response to the reviewers' comments. Your revised manuscript would need to be seen by the reviewers again, but please note that we would not engage them unless their main concerns have been addressed. 

We appreciate that these requests represent a great deal of extra work, and we are willing to relax our standard revision time to allow you 6 months to revise your study. Please email us (plosbiology@plos.org) if you have any questions or concerns, or envision needing a (short) extension.

**IMPORTANT - SUBMITTING YOUR REVISION**

*Resubmission Checklist*

*Published Peer Review*

*PLOS Data Policy*

*Blot and Gel Data Policy*

Sincerely,

Kris

Kris Dickson, Ph.D., (she/her)

Neurosciences Senior Editor/Section Manager

PLOS Biology

kdickson@plos.org

REVIEWS:

Reviewer #1: Contextual fear memory traces are believed to consist of subsets of hippocampal principal neurons that are specifically activated during the process of memory formation, which couples the CS (conditioned stimulus; a specific context here) with US (unconditioned stimulus; footshock here); moreover, the recall of the specific memory requires the re-activation of the same subset of principal neurons. Previous studies indicate that somatostatin (SOM)-expressing interneurons, which receive excitatory inputs from the medial septum and inhibitory inputs from the nucleus incertus (NI), provide a global inhibitory tone (lateral inhibition) for the selective activation of the memory engram (M. Lovett-Barron et al., 2014; Stefanelli et al., 2016). More recently, Nyiri's group reports that coupling the optogenetic stimulation and inhibition of NI GABAergic neurons with US prevented and facilitated the formation of fear memories, respectively (Szőnyi et al., 2019). Here, Zichó et al. from the same group report that coupling the optogenetic inhibition of hippocampal SOM cells or optogenetic activation of NI GABA neurons during memory formation enables future fear memory recall using the same optogenetic manipulation. 

To reconcile the seemingly discrepant observations between this study and the Szőnyi et al., 2019 study, the authors hypothesized that inhibiting a subset of hippocampal SOM neurons or activating NI GABA cells may not only disrupt the association between the context and the footshock (as shown in the Szőnyi et al., 2019 study) but also enable the formation of a new memory that associates the activity change of these neurons with the footshock, and the re-inhibition of SOM neurons or re-activation of NI GABA neurons leads to memory recall (Paragraphs 2-3 in Discussion). In essence, the authors seemed to theorize that the optogenetic disturbance of neuronal activity serves as CS here. I feel that the current study provides some novel observations and raises an intriguing interpretation. However, the authors need to provide additional experimental evidence and improve the quality of current experiments to strengthen their conclusions. 

Major concerns:

(1) One can make two predictions based on the authors' hypothesis. First, the timing of inhibiting hippocampal SOM neurons or activating NI GABA cells does not need to be strictly aligned with footshock. Rather, a range of interval duration between the optogenetic manipulations and US should also support memory formation and recall, whereas such manipulations after US shall be ineffective. Data from this new experiment will be critical for establishing authors' key conclusion. 

(2) Second, the GABAergic input from the NI is not truly special, in that any other GABAergic input to hippocampal SOM neurons will also support similar fear memory formation and recall. The authors may address this concern by using cell type-specific rabies tracing to illustrate the presynaptic partners of DG SOM neurons and then test and discuss the potential roles of those inputs. For example, do some other inhibitory inputs also contribute to memory formation and recall, whereas the excitatory inputs lack such effect?

(3) Several experiments need to be done more rigorously. For example, the authors used c-Fos immunopositivity as indicators of neuron activation in the hippocampus and the NI (Figs. 1 and 5). Given the known limitations of c-Fos labeling, fiber photometry of Ca2+ signals will provide stronger support to the authors claims. In Figs. 3 and 6, the authors used the proximity of FP-labeled NI terminals to hippocampal SOM neurons and NI GABA neurons as indicators of synaptic connections. Slice physiology is required to rigorously establish functional connectivity. 

Minor concerns. 

1. In Fig. 1, the authors show that optogenetic inhibition of DG SOM neurons induces fear recall in a new context (Fig. 1A & B), whereas chemogenetic inhibition prevents fear recall in the conditioned context (Fig. 1H). Does this mean that the chemogenetic inhibition overwrote the inhibition pattern of SOM neurons in association with the conditioned context? Please discuss. 

2. I am also puzzled by the authors' emphasis of "a subset of" DG SOM population in memory recall. Such emphasis doesn't seem justified, since it is impossible to completely silence this cell population in its entirety, and the authors didn't test the effect of inhibiting various proportion of DG SOM neurons. In addition, the authors mentioned that "only about 55% of SOM cells were infected by the AAVs below the tip of the optic fibers" (Fig. S1G), but this figure panel did not show such data. A more representative image and group data are needed.

3. To directly test the role of DG SOM neurons in mediating the effects of NI GABA neurons, the authors should activate NI GABA neurons and simultaneously block the GABA receptors of DG SOM neurons (or simultaneously activate these neurons). I recommend the authors to perform this experiment. At the very least, they should discuss the limitation of their current dataset. 

4. The authors should provide physiological data to directly confirm that expressing eOPN3- or ArchT effectively mediates optogenetic inhibition. In addition, I am curious about the reason behind using two different optogenetic inhibition tools in a same study. 

5. AAV2/1 vectors have been used as an anterograde trans-synaptic tracing tool. Did the authors observe the labeling of DG GCs following the infection of SOM neurons with AAV2/1-hSyn-SIO-eOPN3-mScarlet virus? 

6. Data from the authors' group (Szőnyi et al., 2019) show that the activity of NI neurons is closely associated with locomotion. In addition, another study shows that activating NI neuromedin B neurons, which consist of mainly GABA cells, enhance animal arousal and locomotion (Lu et al. 2020). Did the authors observe any locomotor activity change following optogenetic manipulations of NI GABA neurons? 

7. NI GABA neurons project to many brain areas, including the medial septum, the interpeduncular nucleus, and the median raphe, all of which are closely associated with hippocampal circuits. Stimulating NI GABA neurons or their axonal terminals will lead to the release of GABA and peptide co-transmitters in these brain areas, either directly or through antidromic activation. Therefore, some of the behavioral effects may not result from direct inhibition of hippocampal SOM neurons. The authors should discuss the caveat of their methods. 

8. The Result section can be better organized. There are too many subsections. Some figure panels, including those related to PV neurons, could be moved to the supplementary figures. 

Reviewer #2: Zichó et al., 2022 (Plos Biology)

In this manuscript, Zichó et al expand on previous findings from the same group (Szonyi et al., 2019) and investigate the effect of inhibitory networks in inducing fear memories, with special focus on SOM hilar neurons and nucleus incertus (NI) inputs. In summary, the authors show that inhibitory neurons from the hippocampus and NI contribute to reinstate patterns of activity that are important for memory expression. Non-specific manipulation of those same inhibitory networks, however, impair memory expression. This is an interesting, well executed body of work that benefits from an intelligent design and the strength of optogenetics. The paper is well written, figures are clear, information can be easily found and the amount of work is considerable. Still, the flow of the results is at times difficult to follow because little interpretation linking the results or rationale of using different experiments such as the use of different contexts, inhibition vs. activation, etc, is given. The reader gets easily lost.

In sum, I would accept the manuscript after knowing the opinion of the authors to the following major and minor questions.

Major: 

1) In a more conceptual level, I have problems understanding the significance of the inhibition experiments, and how it translates to the more widely accepted notions of memory engrams. The introduction makes a big point that no real mechanism for precise re-activation of a given memory engram is known, but the manuscript fails to address this since reactivation of the original engram is never evaluated upon interneuron optogenetic manipulations. This would require a new battery of experiments in TRAP2 or TetTag mice that are probably too much to ask in the context of this revision. Nevertheless, this shortcoming should be properly addressed in the discussion. For example, in figure 1, the results of the opto-experiment are quite intriguing but reactivation of the original ensemble of engram cells in the DG is not assessed. Re-inhibition of SOM cells could lead to a precise reactivation of the original ensemble as the authors suggest, but it could also increase the mere probability of chance reactivation just by leading to the recruitment of larger ensembles of cells, as the authors actually show. 

2) In several panels (eg, Fig 2B, 3H, 4B, ...) mice show 0% freezing. This is rather unusual, and can heavily skew the results. 

a. Figure 2. It is weird to observe that there is basically no freezing in eOPN3 mice on the first, light-off period of the 8th day. This contrasts with the rest of experiments in the manuscript, especially with the previous experiment where the authors even discussed that "mice showed some fear behaviour because it was not possible to build a novel environment that is completely different from environment B". Can the authors explain? Could this be a general "encoding error" effect derived from the inhibition of PV neurons during shock delivery? Were the same mice tested in the environment B afterwards as in the first SOM experiment?

b. Figure 3/4: Once more, the lack of freezing on the readout day, in this case in control animals, is worrying. Is there any consistent explanation for this?

3) Figures 3 and 4 should be combined. 

Minor:

Fig1:

The DG-SOM inhibited group ("associated" mice) does not show recall impairments in the same context (day 9), which seems at odds with similar experiments done in CA1 (Lovett-Barron et al., 2014, Szonyi et al., 2019). This needs to be explained. Was opto-silencing done also in the same context (see my comment on chemo-opto manipulations below)?

In 1A, 8th day the initial light off is indicated as a 3min period whereas in the text is stated 2min.

In 1B, was there a batch effect? Were all mice run at the same time? 

Optic fibers are placed on top of the DG, whereas SOM neurons are located mostly in the hilus. Is there any way to know to which extent the light reaches the hilus and SOM neurons are inhibited? Can c-Fos be quantified within SOM cells as in exp 1G? Related to this, what is the evidence that eOPN3 is actually working?

I have troubles putting the opto and chemo-genetic results together, since they seem contradictory: Re-inhibiting SOM via optogenetics induces recall in a neutral context, but inhibiting SOM via chemogenetics reduces freezing in the same context. In both cases the expected effect in DG engram cells should be the same, with the difference that in the chemogenetic experiment the silencing is not specifically targeted to the "encoding" SOM cells… although they are most likely inhibited as well. Is that not contradictory? Nonetheless, to ascertain if this was an artifact derived from the method used and to highlight the specificity that authors seek it would have been interesting to inhibit the SOM neurons in the conditioned context via optogenetics in day 9 of the first experiment. 

Fig. 2:

Merging A-D together with the arrangement of F-G makes the figure a bit confusing. Rather than making a new figure I would consider re-arrangements between panels if the authors agree. 

F. As this experiment is introduced, I understood that the goal was to test whether the memory trace of a contextual memory is lost or silent upon inhibition of CA1 SOM cells during conditioning, which was linked to decreased freezing (at least via manipulations of NI fibers) in their previous work (Szonyi paper). Re-inhibition increases freezing but this does not necessarily mean that the same memory trace has been reactivated. Once more, engram experiments would be necessary to make such claims. 

Is there any reason to use preferentially eOPN3 or ArchT in these different experiments? 

Fig. 3

Why 40 minutes for fos expression when in an earlier experiment 60 min were used?

Was it investigated whether NI GABAergic fibers also target PV cells? Although the effect of NI fibers on triggering activity in DG GCSs and recall upon re-activation is convincing, whether or not such effect is exclusively due to their action on SOM cells is not shown, and therefore the title of the section in the text is not correct. This specificity could be addressed via simultaneous manipulation of NI fibers and SOM hilar cells on the 8th day, or by showing the effect of NI fiber manipulation on SOM activation with fos/IEGs stainings. Regardless, the effect of NI fibers on PV populations should be assessed. 

Figure 3B would benefit from a zoomed-out view. 

Fig. 4

Minor: in the figure baseline and shock delivery occurs in the same, 6th day, whereas in the text is stated "the next day". Which of the two? 

Fig. 5

The order of the panels is again confusing. I wonder if the NI fiber and cell bodies manipulations could be merged in the same figure.

Freezing levels at remote time are very high compared to the rest of the paper, which is the interpretation for this? What is the correlation of fos in the DG vs. vGAT fos in the NI at remote times?

5I: How is the 800% delta freezing brought about?

Having the inhibition of NI cells after the recent/remote experiment and correlations seems odd, why not putting it next to the optogenetic activation of the previous figure? 

If here the effect of inhibition is tested via opto- instead of chemo-genetics, why testing it in consecutive days and not in a 2-3-2 min fashion as in the rest of the paper? Also, it is not clear from the text whether the effect of optogenetic inhibition was additionally tested in the different environment B. In other words, could the manipulation of NI cells that were not active during conditioning affect fear in the different context?

Fig. 6

This is a beautiful figure, but it doesn't add much to the story. For the amplified insets showing contacts between cortical fibers and NI INs it would be necessary to include xy projections next to the confocal images (same for fig 3B), and quantifications. Based on these grounds, this figure should be moved to supplement. 

Discussion

I particularly enjoyed the clinical part of the discussion, but figure S9 is not visually pleasing. Personally, I think that including small bits of discussion/interpretation at the end of each results section and not only in the discussion would help navigate the manuscript, but I leave this to the discretion of the authors. 

Other minor comments:

- Page 2, line 21: Han et al. 2007 did not investigate excitability

- Page 5, second paragraph: The fact that these experiments were run in SOM-Cre animals is missing from the text

- Page 9, line 5: What environmental stimuli?

- Page 10, line 3: Something cannot be "very specific". It's either specific or not. 

Reviewer #3: This is an interesting and impactful study, focused on control of memory encoding and recall by hippocampal somatostatin-expressing interneurons, and the NI neurons that regulate their activity. There is much to praise and little to criticize in the study. Aside from a few recommendations for improving the text and figures, I have no major concerns that would slow publication. These recommendations are described below:

-Abstract 

Sentence 1 "Fear-related memory traces are encoded by sparse populations of hippocampal principal neurons that are recruited based on their inhibitory-excitatory balance during memory formation"

On this and some other occasions the authors state fear-related memory, or simply fear memory, and the involvement of the hippocampus. It might be better to change "Fear-related memory traces" to "contextual fear memory traces" or "spatial and contextual cues of fear-related memory traces"

- Introduction, page 1, line 21 - Cai et al 2016 should also be cited here

- Introduction, page 1, line 23 - "recall.." should be changed to "consolidation and recall" Delorme et al 2021 and Raven et al 2021 should be cited

- Introduction, page 1, lines 31-33 - and also modulate principal cell reactivation during consolidation (Delorme et al, Raven et al 2021)

- Results, page 3, line 27 - Cai et al 2016 should also be cited here

- Results, page 5, lines 18-19 - this is a question to address in the discussion section, I believe, which is: What would happen if CNO were given during both training and testing?

- Discussion, 1st 2 paragraphs - here, again, it would be nice to see some discussion of the recently-described role of SOM interneurons in fear memory consolidation (Delorme et al 2021)

- Figure 1b - Should check p-value for 1B non-associated group. There seems to be a trend that laser stimulation (SOM inhibition) during the test impairs memory? Or could this be a time-in-box effect? Possibly the only way to know is light on after a second time light off? So OFF-ON-OFF-ON?

- Figures 2 and 5 - it would be nice if these panels were laid out differently, as presented it is hard to follow - i.e., having experimental data described later in the text presented above data described earlier

---

## [Decision Letter · Decision Letter 2]

3 May 2023

Dear Dr Nyiri,

Thank you for your patience while we considered your revised manuscript "Fear memory recall via hippocampal somatostatin interneurons" for publication as a Research Article at PLOS Biology. This revised version of your manuscript has been evaluated by the PLOS Biology editors, the Academic Editor and the original reviewers.

Based on the reviews, we are likely to accept this manuscript for publication, provided you satisfactorily address the remaining points raised by the reviewers and the following data and other policy-related requests.

a) Please change your title so that it includes an active verb - we suggest "Fear memory recall involves hippocampal somatostatin interneurons."

b) Please address the remaining requests from reviewers #1 and #2.

c) Please mention the study species in the Abstract.

d) Please address my Data Policy requests below; specifically, we need you to supply the numerical values underlying Figs 1BCDGI, 2BCDFGH, 3EGHI, 4BCDFG, 5DEFHI, 6OPQ, S1ABCDE, S2BCDEF, S4BC, S5ABCD, S6ACDF, S7B, S8, S9, either as a supplementary data file or as a permanent DOI’d deposition.

e) Please cite the location of the data clearly in all relevant main and supplementary Figure legends, e.g. “The data underlying this Figure can be found in S1 Data” or “The data underlying this Figure can be found in https://doi.org/10.5281/zenodo.XXXXX”

f) We note that the Methods are in the Supplementary Information; please incorporate them into the main manuscript file.

We expect to receive your revised manuscript within two weeks. 

*Published Peer Review History*

*Press*

Sincerely,

Roli Roberts

Roland Roberts, PhD

Senior Editor,

rroberts@plos.org,

PLOS Biology

DATA POLICY:

Regardless of the method selected, please ensure that you provide the individual numerical values that underlie the summary data displayed in the following figure panels as they are essential for readers to assess your analysis and to reproduce it: Figs 1BCDGI, 2BCDFGH, 3EGHI, 4BCDFG, 5DEFHI, 6OPQ, S1ABCDE, S2BCDEF, S4BC, S5ABCD, S6ACDF, S7B, S8, S9. NOTE: the numerical data provided should include all replicates AND the way in which the plotted mean and errors were derived (it should not present only the mean/average values).

SPECIES INDICATED IN THE ABSTRACT? 

- Please note that per journal policy, the model system/species studied should be clearly stated in the abstract of your manuscript. 

DATA NOT SHOWN?

Reviewer remarks:

Reviewer's Responses to Questions

PLOS authors have the option to publish the peer review history of their article (what does this mean?). If published, this will include your full peer review and any attached files.

Reviewer #1: Yes: Minmin Luo

Reviewer #2: No

Reviewer #3: No

Reviewer #1:

[identifies himself as Minmin Luo]

I appreciate the authors' efforts of addressing my concerns. They have provided additional experimental evidence to support the conclusion that hippocampal SOM neurons and NI GABA neurons participate in memory formation and recall. More specifically, they now show that aligning optogenetic stimulation with aversive stimuli produces more effective association (Fig. S2A-C). They ephys recordings in vivo now more convincingly demonstrate the functional connectivity of the ACC-NI-DG pathway (Figs. 3F-I, 6N-Q, and S4). The revisions in Discussion are also appropriate. 

I only have one minor concern. The retrograde tracing data shown in Fig. S3C illustrate only a small number of labeled neurons in the NI (one or two cells per section). It will be helpful if the authors could clarify.

Overall, the manuscript has been substantially strengthened by the revisions. I am happy to support its publication at Plos Biology. 

Reviewer #2:

Round 2 revision Zicho et al.

I thank the authors for their detailed responses and new additions to the manuscript. In vivo electrophysiology complements well the results on NI to DG SOM connectivity. I agree with some of the responses from the authors while finding others no entirely convincing. Specifically, I would like to make the following final comments:

I believe the align vs. shift experiment is interesting and well-conceived. Unfortunately, the increase in freezing in B, although significant, is difficult to interpret biologically (from 2.6 to 6.9 %) and falls far from the bigger effect observed in the first experiment, in which additionally a different opsin had been used. These discrepancies should be explained or at least justified in the text. 

I completely agree with the response of the authors to my question about "increasing chance reactivation alone". Indeed, the non-associated mice are the perfect control for this, which I overlooked. Stressing this point, maybe when talking about figure 1D, could support the text. 

Evidence that the optogenetic approaches effectively inhibit target cells is still missing, even though it was asked by me and reviewer 1. Previous works demonstrating its effectiveness supports its choice, but methodological differences might lead to different results. This requires further attention. 

It would be desirable to adopt a common formatting for the graphs and data depiction (e.g., showing or not individual data points in the bar graphs). Additionally, I and other reviewers commented on the strange arrangement of figures (such as fig2 and 5), but these were disregarded. I leave this to the discretion of the authors or the editors. 

Regarding my question about differences in the effect of SOM-inhibition in CA1 and DG at recall, I find the answer of the authors to be a bit superficial ("caused by inherent differences between CA1 and DG subnetworks") and inaccurate (SOM inhibition in CA1 would probably also increase activity in this area). Anyway, the conclusion for the manuscript does not change.

More generally, and concerning the response letter, I would advise strongly against overinterpreting beyond what is strictly shown by the data. The experiments do show the effect of re-inhibition in memory encoding and recall, but in their response the authors state: "This is partly because the well-defined population of cells that we manipulated are synaptically tied to their principal neurons. As a result, they had changed the excitability of a similar population of principal cells the same way, both during association and during recall. Because we modulated strictly the same cells during both association and recall, therefore these powerful inhibitory cells shaped the excitability and therefore the selection of these cells associated with the given events the same way". While I agree with the feasibility of this interpretation, strictly speaking, this was never shown in the present study. Once more, engram studies and/or slice physiology of principal cells during optogenetic inhibition of SOM cells would be necessary for this (which is probably out of the scope of the present manuscript, and not needed). Something similar occurs when the authors repeatedly refer to "patterns of activity" that are lost or reinstated after chemo- or opto-inhibition, respectively, because such "patterns" were never measured and are hardly reproducible with opto-protocols. The results are consistent and solid, but interpretation can go in different directions and authors should tone down some of their statements. 

Reviewer #3:

The authors have done an outstanding job of responding to prior concerns. The present study will shape our thinking in the field for years to come, and they should be congratulated on this nice work.

---

## [Editor Report · Decision Letter 3]

9 May 2023

Dear Dr Nyiri,

Thank you for the submission of your revised Research Article "Fear memory recall involves hippocampal somatostatin interneurons" for publication in PLOS Biology. On behalf of my colleagues and the Academic Editor, Jozsef Csicsvari, I'm pleased to say that we can in principle accept your manuscript for publication, provided you address any remaining formatting and reporting issues. These will be detailed in an email you should receive within 2-3 business days from our colleagues in the journal operations team; no action is required from you until then. Please note that we will not be able to formally accept your manuscript and schedule it for publication until you have completed any requested changes.

Sincerely, 

Roli Roberts

Senior Editor

PLOS Biology

rroberts@plos.org